# FGFR signaling establishes spatial gradients of secretory cell identities along the airway proximal-distal axis

Alexandros Sountoulidis [1] ✉, Jonas Theelke [1,8], Andreas Liontos[1,8], Alexandra B. Firsova [1], Orane Eliot[1], Janine Koepke[2], Pamela Millar-Büchner[2], Louise Mannerås-Holm[3], Åsa Björklund [4], Athanasios Fysikopoulos [2], Antonia Kelm[1], Eleni Bouloukou[2], Konstantin Gaengel [5], Fredrik Bäckhed[3,6], Christer Betsholtz[5,7], Werner Seeger [2], Saverio Bellusci[2] & Christos Samakovlis [1,2] ✉

Secretory cells are major structural and functional constituents of the lung airways. Their heterogeneity, spatial organization and specification mechanisms are partially understood. Here, we analyze secretory lung cell-types at single-cell resolution. In the airway epithelium, we find opposing, partially overlapping gene-expression gradients along the proximal-distal airway axis superimposed on a general gene program encoding detoxification. One graded program is elevated proximally and relates to innate immunity, whereas the other is enriched distally, encoding lipid metabolism and antigen presentation. Intermediately positioned cells express moderate levels of both graded programs creating a differentiation continuum towards each end. Lineage tracing analysis during development reveals the sequential establishment of the gradients in common epithelial progenitors postnatally. We show that Fgfr2b regulates the airway patterning by inducing and maintaining high levels of lipid biosynthesis and vesicle trafficking in distal airways and down-regulating innate-immunity genes in vivo and in airway organoids. Our analysis offers a framework for studying epithelial and lung tissue organization to better understand cellular roles in tissue-level pathology.

The airway epithelial network is a seamless conduit of air to the lung alveoli and provides a first barrier against inhaled pathogens and pollutants[1]. The airways can be anatomically divided into extra-lobar (trachea and two main bronchi) and intra-lobar compartments[2] with distinct cell compositions. Basal cells, for example, are localized in the extra-lobar compartment of the mouse airways[3] and gradually decrease along the proximal-distal (PD) axis of the human intra-lobar airways[4]. The epithelial cells on the airway surface coordinately

[1]Department of Molecular Biosciences, The Wenner-Gren Institute (MBW), Science for Life Laboratory, Stockholm University, Stockholm, Sweden. [2]Department of Internal Medicine, Universities of Giessen and Marburg Lung Center (UGMLC), German Center for Lung Research (DZL), Institute for Lung Health, Cardio-Pulmonary Institute (CPI), Giessen, Germany. [3]The Wallenberg Laboratory, Department of Molecular and Clinical Medicine, Institute of Medicine, Sahlgrenska Academy, University of Gothenburg, Gothenburg, Sweden. [4]Department of Life Sciences, National Bioinformatics Infrastructure Sweden, Science for Life Laboratory, Chalmers University of Technology, Göteborg, Sweden. [5]Department of Immunology, Genetics and Pathology, Rudbeck Laboratory, Uppsala University, Uppsala, Sweden. [6]Novo Nordisk Foundation Microbiome Health Initiative, National Food Institute, Technical University of Denmark, Lyngby, Denmark. [7]Department of Medicine Huddinge, Karolinska Institutet, Huddinge, Sweden. [8]These authors contributed equally: Jonas Theelke, Andreas Liontos. ✉e-mail: alexandros.sountoulidis@su.se; christos.samakovlis@su.se

accomplish distinct functions, creating the mucociliary escalator[5]. Secretory cells contribute to tissue homeostasis by detoxification of inhaled xenobiotics[6] and by secretion of mucins and antibacterial peptides[7–11]. Mucus and inhaled particles are propelled out of the airways by multiciliated cells[12–14]. More recently, additional secretory cell-types were defined by co-expression of various combinations of secretoglobins, mucins and surfactant proteins in distinct regions of the trachea and human airways[15–20].

In homeostasis, airway secretory cells can self-renew and produce ciliated cells[21,22]. Upon injury, however, they can reconstitute both the airway and the alveolar epithelium[5,20,23,24]. Extensive research efforts have described several subsets of airway secretory cells contributing

to tissue repair upon different types of injury. These cells include the variant (v) club cells[25,26], the *Upk3a*[pos] (u) club cells[27] located near neuroendocrine cells (NE), the bronchioalveolar stem cells (BASCs) in terminal bronchioles (TBs)[26,28,29], the β4[pos] CD200[pos] Scgb1a1[pos] cells in distal airways[23] and cells in activated transitional states (ADI[30], DAPT[31], and PATS[32]). The presence of such a large variety of injury-responsive cells suggests that the airway epithelium is highly heterogeneous and capable of adopting regenerative characteristics upon damage. Most secretory cell types emerge postnatally, and our understanding of their lineage relationships and differentiation mechanisms is limited.

Here, we focused on the airway secretory cells during mouse development and homeostasis to capture their topologies,

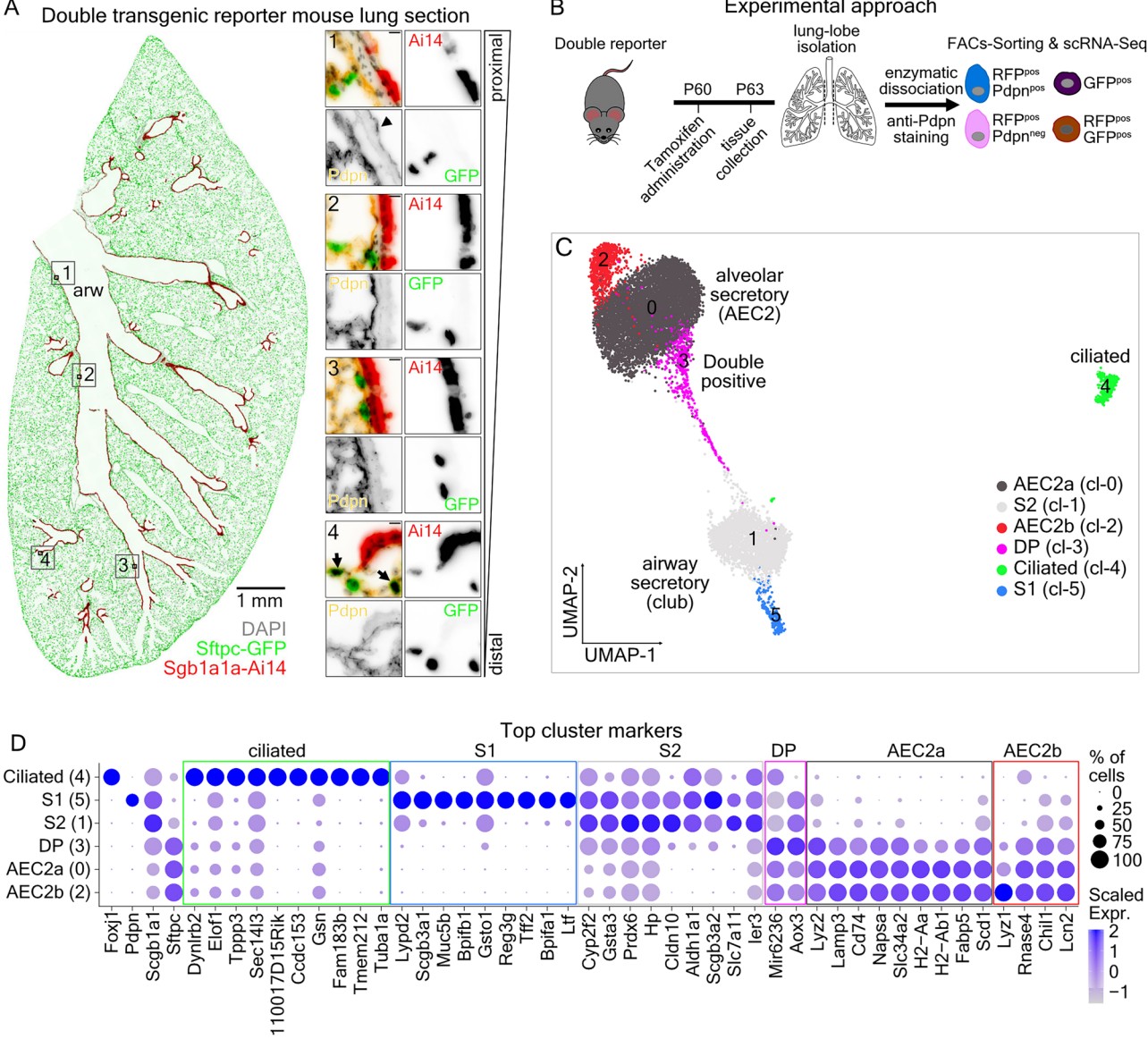

**Fig. 1 | Characterization of lung secretory cell heterogeneity. A** Color-inverted fluorescence image of a left-lung section from an Scgb1a1-CreER[T2pos/neg]; Rosa26-Ai14[pos/neg]; Sftpc-GFP P63 reporter mouse, three days after Tamoxifen administration. DAPI (nuclei): gray, RFP (Ai14): red, GFP (Sftpc): green, Pdpn: yellow. Scale-bar: 1 mm. Inserts correspond to the numbered ROIs. Insert Scale-bars: 20 μm. **B** Experimental outline for the isolation and single-cell RNA sequencing (scRNA-Seq) of the labeled cells from one analyzed reporter mouse. **C** UMAP-plot of 12030 cells from one analyzed mouse, grouped into six clusters. Cluster (cl) 0: AEC2a (alveolar epithelial cell-type-2a) (dark gray), cl-2: AEC2b (red), cl-5: airway secretory cell-type-1 (S1) (blue), cl-1: airway secretory cell-type-2 (S2) (light gray), cl-3:

Scgb1a1[pos] Sftpc[pos] double positive (DP) (magenta) and cl-4: ciliated cells (green). **D** Balloon-plot of known epithelial markers. Ciliated: *Foxj1*, Club cells: *Scgb1a1*, Proximal-airway secretory and AEC1: *Pdpn*, AEC2 and DP-cells: *Sftpc*, in addition to the expression of the top-10 differentially expressed genes of each cluster. The genes were filtered according to average log2 Fold-change (>0.5), Bonferroni adjusted *p*-value (<0.001), and percent of positive cells (>0.25) and the top-10 markers according to average log2 Fold-change were plotted. Gene order follows the cluster order. Balloon size: percent of positive cells. Color intensity: scaled expression (blue: high and gray: low). The results of all statistical analyses and exact *p*-values are provided in Supplementary Data 1.

differentiation trajectories and regulation. Our work reveals a continuum of cell-states along the PD-axis of the airway network. We identified a general airway secretory gene-expression program that relates to detoxification, in addition to at least two complementary graded ones. A gene program encoding innate immunity peaks in proximal cells and gradually decreases distally. An opposing gradient of gene expression programs encoding antigen presentation, lipid and surfactant metabolism peaks in distal cells and decreases proximally. Cell lineaging analysis combined with single-cell RNA sequencing (scRNA-Seq) indicates that the differentiation programs of airway secretory cells become sequentially activated in a common embryonic secretory cell progenitor. We show that postnatal FGFR-signaling is a key regulator of airway cell differentiation in vivo and in organoid cultures in vitro. FGFR is also required in adult homeostasis to maintain graded gene-expression patterns and cell heterogeneity. This role of FGFR, in addition to its crucial function in alveolar secretory type-II cell (AEC2) differentiation and maintenance[33–38], suggests a tissue-wide function for FGFR2b in promoting distinct differentiation states depending on the topological coordinates of lung secretory cells.

## Results

### Analysis of epithelial cell heterogeneity in the adult lung

We first isolated secretory mouse lung cells from a double transgenic reporter strain expressing a green fluorescence protein (GFP) in the alveolar secretory (AEC2) cells (Sftpc-GFP)[39] and a tamoxifen-inducible (tam) red fluorescence protein (RFP)[40] in the airway secretory (club) cells[22] (Scgb1a1creER^T2-Ai14). The cell surface protein Pdpn is selectively expressed apically in secretory cells of the proximal airways (Fig. 1A and ref. 41). Therefore, we additionally used Pdpn antibodies in conjunction with the RFP fluorescence to separate two fractions of Pdpn^pos Ai14^pos and Pdpn^neg Ai14^pos cells.

Three days after induction, we FACS-sorted labeled cells (Fig. 1B and Supplementary Fig. 1A) and generated and analyzed 12030 high-quality cDNA libraries (Supplementary Fig. 1B, C). The UMAP (Uniform Manifold Approximation and Projection) plot[42] and differential expression of marker genes in the clusters were consistent with the FACS-sorting criteria (Supplementary Fig. 1D, E). Clusters were annotated according to positivity for known lung epithelial cell markers (Fig. 1C, D, Supplementary Fig. 1F, and Supplementary Data 1). The small cluster-4 (cl-4) contained Foxj1^pos multiciliated cells found in the Scgb1a1creER^T2-Ai14^pos Pdpn^neg cell sorting fraction (99.8%), suggesting that the modest Scgb1a1 expression (Supplementary Fig. 1G) in ciliated cells was sufficient to induce recombination in a few ciliated cells. As expected, cl-4 cells uniquely expressed genes related to cilia (GO:0044782) (Supplementary Data 2).

The Scgb1a1^pos airway secretory cells were separated into three clusters. Cl-5 contains mainly Scgb1a1-Ai14^pos Pdpn^pos sorted cells that also expressed high levels of Scgb3a1 and Scgb3a2[10] (Fig. 1D). We annotated these cells as S1 (Secretory 1). Cl-1 was almost exclusively (99.2%) composed of Scgb1a1creER^T2-Ai14^pos Pdpn^neg cells (Supplementary Fig. 1D, E), which were also positive for Kdr[43] (Supplementary Fig. 1F) and other, previously reported epithelial cell markers. We annotated these cells as S2. Cl-3 contained 92.4% of the sorted Scgb1a1creER^T2-Ai14^pos Sftpc-GFP^pos double-positive (DP) cells. The remaining cells sorted as DP were evenly distributed among the two alveolar secretory type-2 (AEC2) clusters-0 and -2 (Supplementary Fig. 1D, E), suggesting that some AEC2s also express Scgb1a1. This indicates that Scgb1a1 expression alone does not exclusively identify airway secretory cells[24]. Cl-3 cells likely correspond to the previously described BASCs and did not express any unique marker but moderate levels of both the S2 and AEC2 markers (Fig. 1D), indicating an intermediate cell-state.

The clusters of AEC2 cells (cl-0, AEC2a) and (cl-2, AEC2b) only differed in Lyz1 expression in the AEC2b cluster (Supplementary Fig. 1H), as reported previously[44]. Both alveolar clusters expressed high

levels of genes involved in lipid biosynthesis (GO:0008610) and lipid transport (GO:0006869), in addition to genes implicated in the regulation of leukocyte activation (GO:0002694), such as MHC class-II genes (Supplementary Data 2).

Overall, this analysis confirms previous reports defining two airway-secretory cell-types, a group of Scgb1a1^pos Sftpc^pos cells, two secretory alveolar cell identities that differ in the expression of Lyz1 and a cluster of Foxj1^pos ciliated cells.

### Gene expression patterns suggest distinct secretory cell functions

The continuous arrangement of the lung secretory cells in the UMAP-embedding suggested a spectrum of graded gene expression that creates intermediate cell-states, bridging the main bodies of each of the above clusters. To explore the transcriptional heterogeneity along this continuum, we used diffusion maps[45] and trajectory analysis (Fig. 2A). We ordered equal numbers of randomly selected cells from each cluster and identified 1563 differentially expressed genes (DEGs) along a trajectory that starts from S1 cells and ends at AEC2b. These genes can be grouped into 10 stable modules, according to their expression pattern similarities in the interrogated cells, defining groups of co-expressed genes at the cellular level (Fig. 2B and Supplementary Data 3). The aggregated gene expression scores for each gene-module reflects its expression pattern. Module-3 genes showed equal activation in the part of the trajectory with S1 and S2 cells, but also in ciliated cells (Fig. 2B and Supplementary Fig. 1I), representing a general airway gene-expression program. Modules-5 and -2 were gradually reduced along the trajectory towards AEC2 cells, while modules-5 and -2 were graded in the opposite orientation, decreasing towards S1. Modules -7 and -8 were enriched in the part of the trajectory with mainly S2 cells, but only module-7 was enriched in ciliated cells. Module-9 genes were expressed in all but the S1 cells, and modules-10 and -6 in all but S2 cells. This analysis revealed groups of co-expressed genes with shared, distinct and graded activation patterns along the epithelial secretory cell trajectory.

Gene ontology (GO) analyses for each of the gene-modules suggested that the co-expressed genes in each module underpin common biological functions (Fig. 2C, D and Supplementary Data 3). For example, module-5 includes genes related to innate immunity regulation (GO:0002682, Reg3g[46], Ltf, Bpifb1[47], P2rx4[48], Il13ra1[49], and Mfge8[50]) and their expression is restricted to the part of the trajectory composed of S1-cells only. Module-2 genes are also highly expressed in the S1 trajectory and become gradually decrease towards S2 and DP cells. These genes encode various metabolic enzymes (e.g., Gsta1, Acsl1, and Gstm5), cytokines (Cxcl1, Cxcl2, Cxcl5, and Cxcl17) and interferon-induced antiviral proteins (Ifitm1, Ifitm3, and Ifit1), suggesting differential responses to chemicals (GO:0070887) and viruses (GO:0034097) along the trajectory. This is also supported by previous functional analyses of the module-2 transcription factors (TF) Six1[51] and Spdef[52], which are involved in airway inflammation and by the upregulation of Irf7 in airway secretory cells upon RSV infection[53].

Module-3 genes are uniformly expressed in the part of the trajectory with S1 and S2 cells and primarily encode metabolic and detoxification enzymes (e.g., Cyp2f2, Aldh1a1, and Gsta3), supporting the notion that a central feature of all airway secretory cells is the response to xenobiotics (GO:0009410, GO:0006749) and detoxification of inhaled air.

In the intermediate part of the trajectory, where S2 cells are located, there is a selective enrichment of the module-7 genes, which relate to the general term "cellular development" (GO:0048869). This module contains developmental genes, such as Shh[54–56] and the negative regulators of airway inflammation Kdr, Sema3e, Sema3a, and Nr1d1[43,57–59].

At the end of the trajectory (including mainly DP cells and AEC2s), there is a selective and gradual upregulation of module-1 genes. These

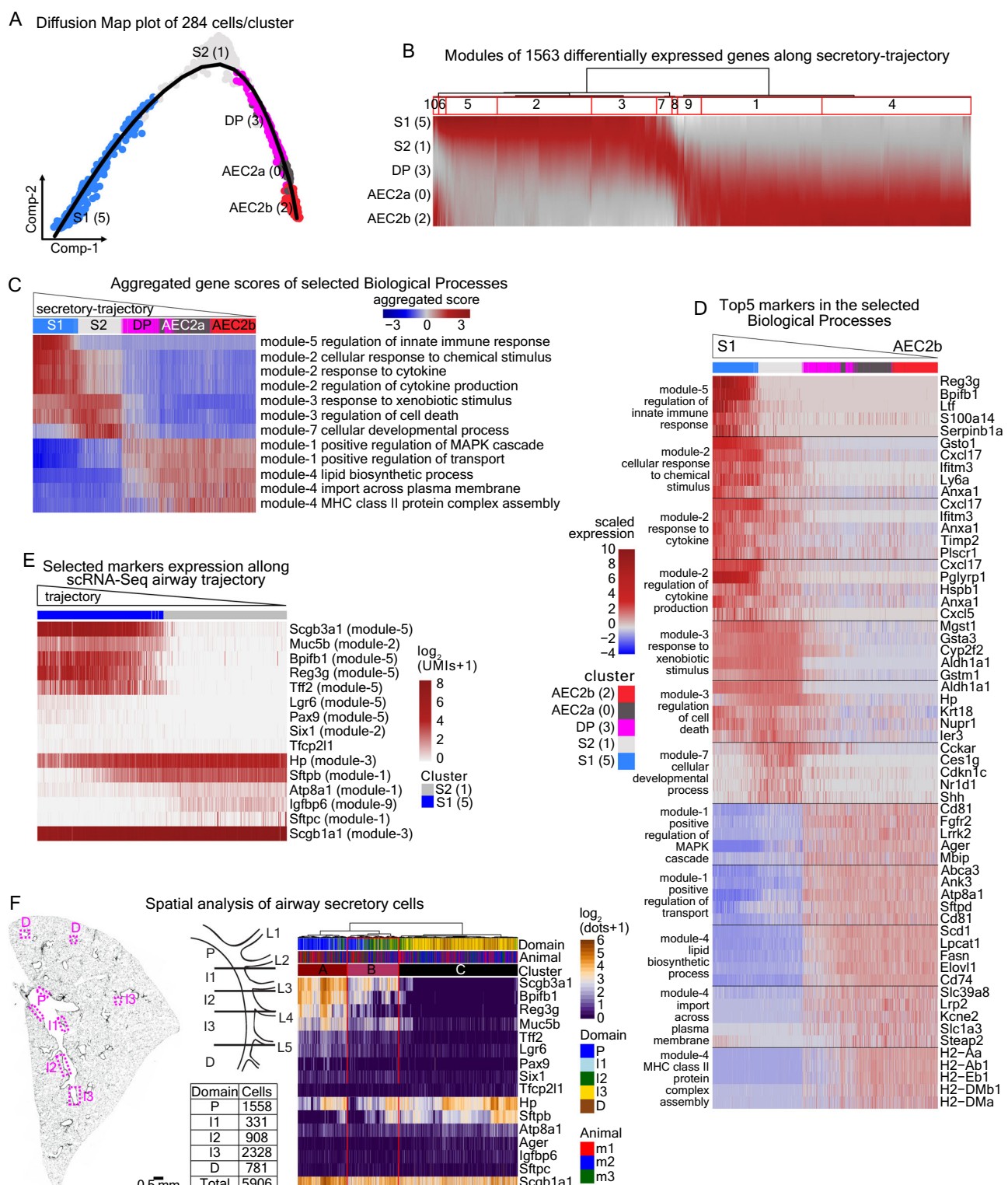

A  Diffusion Map plot of 284 cells/cluster

B  Modules of 1563 differentially expressed genes along secretory-trajectory

C  Aggregated gene scores of selected Biological Processes

D  Top5 markers in the selected Biological Processes

E  Selected markers expression along scRNA-Seq airway trajectory

F  Spatial analysis of airway secretory cells

| Domain | Cells |
|--------|-------|
| P | 1558 |
| I1 | 331 |
| I2 | 908 |
| I3 | 2328 |
| D | 781 |
| Total | 5906 |

encode kinases like the *Fgfr2* and *Lrrk2*, required for maintenance and function of the alveolar type 2 cells[33,34,36,60]. Additionally, module-1 genes include *Atp8a1*, *Abca3* and the Rab-family genes *Rab27a* and *Rab34* relating to phospholipid transport (GO:0051050) and vesicle trafficking, necessary processes in surfactant production and secretion.

Finally, the cells in the two alveolar epithelial clusters upregulate module-4 genes, which encompass known regulators of alveolar cell differentiation and maintenance, like *Etv5*[61] and *Nkx2-1*[62,63], together with genes relating to lipid biosynthesis (GO:0008610) and transport

across the plasma membrane (GO:0098739). Notably, the enriched expression of *H2-Eb1, H2-DMb1, H2-Dma, H2-Aa,* and *H2-Ab1* involved in MHC class-II complex assembly (GO:0002399) suggests a selective role of alveolar secretory cells in antigen-presentation to immune cells.

This analysis identifies several shared but also differentially activated gene expression programs in lung secretory cells. The expression intensity of the corresponding genes along a continuum of cell-states suggests that prominent biological processes of the lung epithelium relating to immune responses, detoxification, lipid biosynthesis and ion transport are segregated in the airway and alveolar

**Fig. 2 | scRNA-Seq trajectory recapitulates the airway proximal-distal patterning. A** Diffusion-map of secretory cell clusters. Colors and numbers as in Fig. 1C. Line: estimated pseudotime-trajectory by Slingshot. (**B**) Heatmap of the 1563 differentially expressed genes (FDR < 0.001 and meanLogFC > 1) along pseudotime, based on tradeSeq. AssociationTest of TradeSeq was used for unadjusted *p*-value calculation, and FDR was estimated with Benjamini & Hochberg correction. The dendrogram of hierarchical clustering (left) indicates 10 stable gene-modules. Bootstrapping values: module-1: 0.65, module-2: 0.62, module-3: 0.61, module-4: 0.74, module-5: 0.68, module-6: 0.75, module-7: 0.8, module-8: 0.7, module-9: 0.64, module-10: 0.9. Color intensity: scaled expression. Dark red: high, Gray: low. **C** Heatmap of the aggregated expression scores of the genes in the indicated biological processes (Supplementary Data 3). The number after "m" indicates the module containing the genes in (**B**). Cells were ordered according to pseudotime (Fig. 2A). Red: high, blue: low. **D** Heatmap of the top-5 genes (according to "wald-Stat" score in AssociationTest of TradeSeq) of the indicated biological processes in

(**C**). Cells were ordered according to pseudotime. Color: scaled expression (red: high, blue: low). **E** Heatmap of the selected S1 (cluster-5) and S2 (cluster-1) markers in scRNA-Seq dataset, ordered along pseudotime. Expression levels: $\log_2$(normalized UMI-counts+1) (library size was normalized to 10.000). **F** Left panel: (left) Representative adult mouse lung section stained with DAPI (gray) showing examples of imaged areas along the PD-axis. (Right-up) Cartoon of airway domain classification approach. (Right-bottom) Synopsis of analyzed cell-ROIs from three animals, for the indicated domains (*n* = 3). Right panel: Heatmap of 3096 analyzed airway secretory cell-ROIs, showing the $\log_2$(SCRINSHOT dots +1) signal for the selected markers. Cell-ROI ordering is based on hierarchical clustering. Annotation bars show the (i) airway domains of the cell-ROIs, (ii) the analyzed mouse, and (iii) the indicated cluster. Cluster-A: P-domain cells 38.61 ± 6.89%, I1-domain cells 21.61 ± 2.46%. Cluster-B: I2-domain cells 37.79 ± 7.54%. Cluster-C: I3-domain cells 52.08 ± 6.57%, D-domain cells 69.49 ± 2.20%. Source data are provided as a Source Data file.

compartments but are also expressed in a continuous, graded fashion along that trajectory. Cells in intermediate positions of the trajectory express higher levels of module-7 genes, involved in "cellular development," together with low levels of genes in the remaining modules.

## Spatial single-cell analysis of airway gene programs

To further investigate the airway gene expression gradients, we focused on the spatial analysis of representative marker genes of the modules. We first selected a panel of 18 DEGs (Fig. 2E) and detected their transcripts in situ by SCRINSHOT[64]. We quantified the signals in 5906 manually segmented airway cell-ROIs in distinct positions, based on the stereotyped branching pattern in the left lung-lobe[2] (Fig. 2F and Supplementary Fig. 2A). Hierarchical clustering of the 3096 secretory cells showed high expression of the module-5 genes *Scgb3a1* and *Muc5b* in the proximal domains (P and I1), whereas the module-1 genes *Sftpb* and *Atp8a1* were more abundantly expressed in the distal domains I3 and D, in addition to their expression in alveolar AEC2s. Interestingly, cells in the I2 domain co-expressed moderate levels of both module-1 and module-5 markers (Fig. 2F), resembling the intersection area of the two opposing gradients of module-1 and -5 genes along the secretory cell trajectory (Fig. 2E). This suggests that the S1 cells are located at proximal airways and AEC2s in the alveoli (distally), expressing distinct sets of marker genes at high levels, while cells in between represent intermediate states expressing moderate levels of both types of markers. To further test this hypothesis, we analyzed by immunofluorescence co-staining a few S1 and S2 markers (Scgb3a1, Tff2, Muc5b, Hp, and Atp8a1) relative to E-cadherin, which is homogeneously expressed in the epithelium (Supplementary Fig. 2B, C). This analysis confirmed the RNA expression gradients detected by SCRINSHOT. Importantly, both mRNA and protein quantitative analyses showed that the gradients refer to the expression levels of the interrogated markers and not the frequency of highly positive cells.

Many of the module-2 and -5 genes relate to innate immunity and showed the highest expression levels in proximal airways of the adult lung. We, therefore, examined whether external microbes or the lung microbiome sets these basal expression patterns. We selected antibodies against Scgb3a1 and Muc5b (proximal), in addition to Hp and Atp8a1 (distal), to determine their target protein levels, relative to E-Cadherin in germ-free and pathogen-free conditions. The expression levels of the S1 and S2 markers showed similar distributions regardless of the different environmental exposures (Fig. 3A–D). This suggests that the identified gene expression programs of epithelial secretory cells are specified genetically, but can be further activated upon infections, tissue damage or inflammatory diseases[8]. The good correlation between the graded expression values in the spatial analysis (Fig. 2F) and in the scRNA-Seq diffusion maps (Fig. 2E) indicates that the gene expression trajectory reflects the PD-patterning of gene expression in the airway secretory epithelium (Fig. 3E). The expression

values may thus provide a conceptual ruler for positioning secretory epithelial states along the branches of the airway network. The unchanged marker expression gradients in germ-free mouse lungs suggest that airway cell patterning can be influenced but is not dependent on microbes.

## Adult stem cells follow the general airway patterning rules

We further examined the spatial expression patterns of module markers in rare secretory cell identities like the double positive (DP) Scgb1a1[pos] Sftpc[pos] cells, which have been defined as BASCs because of their higher abundance in bronchioalveolar duct junctions (BADJs)[26,28,29]. We used an additional SCRINSHOT probe panel of 16 DEGs along the secretory trajectory, together with the characteristic NE-cell markers, *Ascl1* and *Calca*. We confirmed the previously reported localization of DP cells in BADJs and close to neuroepithelial bodies (NEBs)[65] (Fig. 3F, Supplementary Fig. 2D and Supplementary Data 4). We found them mainly at airway terminal bronchioles (TB) (42.3%) and to a lesser extent in the alveolar compartment (12.5%) close to BADJs. A small fraction of DP cells (7.1%) was close to NE cells, within a 20 μm radius surrounding the NEB borders. Cells in the airway part of the BADJs express higher levels of the airway (module-3) than alveolar enriched markers (module-1 and -4), whereas cell-ROIs in the alveolar part of the BADJs and in alveoli showed the opposite pattern. Immunofluorescence detection of a few protein markers confirmed the SCRINSHOT results (Supplementary Fig. 2E). The DP-cells in the vicinity of NEBs may correspond to v-club cells[25,26] and were found more distantly from NE cells than the *Upk3a[pos]* u-club cells[27] (Supplementary Fig. 2F). We conclude that even rare secretory cell-types follow the general PD-patterning rule of the lung epithelial network, expressing graded levels of airway and alveolar cell markers depending on their position. Their distribution and distinct identities may reflect unique signals from the NEB microenvironment, such as Notch-signaling[66].

## Airway epithelial cells mature postnatally

To identify when the different airway cell-states are specified and to determine their potential lineage relationships, we labeled embryonic Scgb1a1[pos] cells with a farnesylated GFP variant (Scgb1a1creER[T2]-fGFP)[22] and lineage-traced them. We induced recombination at the onset of Scgb1a1 expression[22], on embryonic day (E) 16 and FACS-sorted GFP-labeled progeny cells at E19.5 and postnatal days (P) P2, P21, and P60 (Fig. 4A) before preparing libraries for scRNA-Seq analysis. We analyzed 354 full-length scRNA-Seq libraries[67] (Supplementary Fig. 3A) using diffusion maps and trajectory analysis and found four distinct trajectories stemming from cl-0, predominantly composed of immature E19.5 and P2 cells. The trajectories ended into four mature clusters containing cells from both P21 and P60 lungs (Fig. 4B and Supplementary Fig. 3B). According to differential expression analysis and previous knowledge, cl-3 corresponds to DP, cl-1 to S2, cl-2 to S1, and

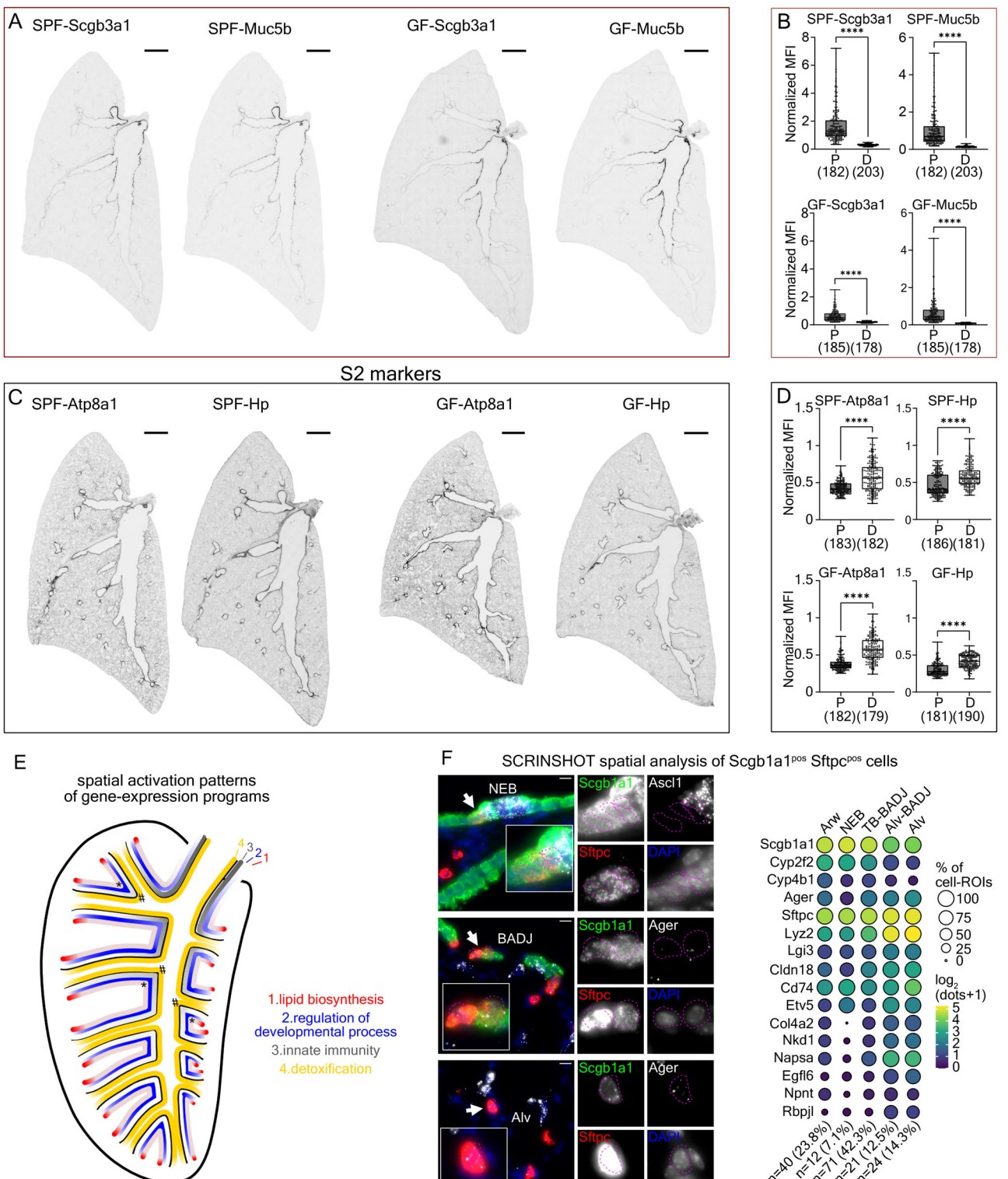

cl-4 to ciliated cells (Supplementary Fig. 3C and Supplementary Data 5) as in the adult-cell dataset. The enriched gene sets of the mature cell clusters largely overlap with those of the adult dataset. Interestingly, the cl-0 of immature airway cells express high levels of *Upk3a Krt15*[46], *Cccnd2,* the WNT receptor *Fzd1, Itm2a*[68] and the AEC1 marker *Ager*[69] (Fig. 4C, Supplementary Data 5). Several of these genes have been also implicated in injury repair[27,70,71]. Next, we used GO-analysis to identify enriched biological processes in each of the clusters (Supplementary Fig. 3D and Supplementary Data 6) and scored the cells along the trajectories based on the expression of the genes corresponding to GO

terms. We found that the perinatal immature cells highly and transiently express ribosomal genes, indicating high levels of mRNA translation (GO:0006412) and ribosome biogenesis (Fig. 4D, Supplementary Fig. 3D, and Supplementary Data 6). The mature airway secretory cell gene-modules, encoding related genes to detoxification, oxidative stress responses, xenobiotic and lipid metabolism become gradually established along the S1 and S2 trajectories but not in DP-cells (Fig. 4D). This is represented by the expression levels of genes encoding characteristic enzymes, such as the *Aldh1a1, Fmo2,* and *Gsta3* (Supplementary Fig. 3E). Interestingly, the representative genes of

**Fig. 3 | Gene expression patterns along the airway PD-axis.**
**A** Immunofluorescence stainings of whole lung sections of specific pathogen free (SPF) and germ-free (GF) 2 months-old mice for Scgb3a1 and Muc5b S1-markers. Scale-bars: 1000 μm. **B** Min to Max box-plot showing all points from the quantification of immunofluorescence mean fluorescence intensity (MFI) of the indicated markers, normalized to the E-Cadherin signal. Numbers in parentheses: number of analyzed proximal and distal cell-ROIs. Statistics with unpaired two-tailed Student's *t* test. The exact *p*-values in (**B**) are SPF-Scgb3a1: <0.0001, SPF-Muc5b: <0.0001, GF-Scgb3a1: <0.0001 and GF-Muc5b: <0.0001. **C, D** As in (**A, B**) for the Hp and Atp8a1 markers that are highly expressed in distal airways. The exact *p*-values in (**D**) are SPF-Atp8a1: <0.0001, SPF-Hp: <0.0001, GF-Atp8a1: <0.0001 and GF-Hp: <0.0001. **E** Graphical representation of the activated gene expression programs (as in Fig. 2C) along the proximal-distal axis of the adult mouse lung airways. Color intensity: activation level. Dark: high, Fade: low. Exceptions in the expression of the lipid metabolism (asterisk) and detoxification (hash) programs relating to neuroepithelial body topology. **F** (Left) SCRINSHOT analysis images for Scgb1a1pos Sftpcpos DP-cells (arrows) close to neuroendocrine (NE) cells (upper panel), terminal bronchioles (TB) (middle panel) and alveoli (lower panel). Magenta dotted-lines: outlines of 2 μm-expanded Scgb1a1pos Sftpcpos nuclei. Arrows: Scgb1a1pos Sftpcpos cells. Sftpc: red, Scgb1a1: green, Ascl1: gray and DAPI: blue. (Right) Balloon plot of the 16 analyzed genes (module-3: *Scgb1a1, Cyp2f2,* and *Cyp4*b1, module-1: *Ager, Sftpc,* and *Lyz2* and module-4: *Lgi3, Cldn18, Cd74, Etv5, Col4a2, Nkd1, Napsa, Egfl6, Npnt,* and *Rbpjl*) in the 168 identified Scgb1a1pos Sftpcpos cells, according to their position. Balloon size: percentage of positive cells. Color intensity: log$_2$(SCRINSHOT dots +1). Yellow: high, Dark blue: low. "*n*": number of cells in the specified position. Source data are provided as a Source Data file.

---

innate immunity (GO:0002682) *Scgb3a1, Tff2,* and *Muc5b* reached high expression only at the very end of the S1-trajectory (Supplementary Fig. 3F-G). The cells along the DP-trajectory gradually increased their ability for lymphocyte-mediated immunity (GO:0002449), upregulating the *Cd74, Ctsc, Hc, Emp2, H2-Aa,* and *H2-Ab1.* The middle part of the DP-trajectory contains perinatal cells that likely contribute to the local extracellular matrix (ECM) organization (GO:0030198), expressing high levels of *Col4a2, Spock2,* and *Matn4* (Supplementary Fig. 3H).

In summary, we found that different clusters of adult airway secretory cells derive from a common embryonic *Scgb1a1pos* progenitor population. Differentiating cells mature postnatally, acquiring their functional characteristics during the first three postnatal weeks (Fig. 4E). Overall, the perinatal airway epithelium shows high ribosomal biogenesis, gradually decreasing over time. The gene programs of innate immunity (S1-trajectory) are established later than those involved in xenobiotic metabolism and reduction of reactive lipid aldehydes (S1- and S2- trajectories), suggesting that cell specification programs are activated sequentially. In the developing distal lung, the differentiating DP-cells transiently contribute to ECM composition and gradually acquire the expression of antigen presentation genes, which are also expressed by the adult AEC2s.

## Graded expression of Fgfr2 along the airways

To define potential regulators of gene expression heterogeneity in airway cells, we interrogated the expression of ligands and receptors that may influence gene expression. The Fibroblast growth factor (FGF) signaling is crucial for lung epithelial branching[72] and is later required for AEC2 differentiation and maintenance[33–38]. We noticed that *Fgfr2* belongs to gene-module-1 of the trajectory analysis (Supplementary Data 3) and is predicted to contribute to the positive regulation of MAPK cascade in the adult lung (Fig. 2C, D). *Fgfr2* is expressed highly in the part of the trajectory including AEC2 and DP cells, gradually decreasing in S2 and S1 airway cells (Supplementary Fig. 4A). This suggested that signaling is not restricted to alveoli but is also evident along the airway network. We further detected *Fgfr2* expression in the perinatal airway secretory cells (Supplementary Fig. 4B) and differentially localized Fgfr2 protein by immunofluorescence (Supplementary Fig. 4C) in P2 lung sections. Co-staining with a Fgfr2β(IIIb)-Fc chimeric protein, to detect the spatial distribution of Fgfr2-ligands together with Fgfr2, showed a punctate staining for the ligand(s), which was higher around the TBs and distal airway epithelial cells and lower at proximal airways. This suggests the potential of more robust pathway activation in the distal airway regions, in addition to the well characterized role of FGFR-signaling in AEC2 differentiation and maintenance[38].

## Perinatal Fgfr2 inactivation disrupts airway epithelial patterning

To examine if Fgfr2-signaling has any role in the establishment of the gene-expression gradients along the airway epithelium, we deactivated the receptor just after birth, when these programs become established. We induced tamoxifen-mediated Fgfr2-inactivation[73] in the *Scgb1a1* cells (Scgb1a1creER$^{T2}$-Fgfr2KO) and utilized Rosa26R-Ai14 (RFP) expression[40] to detect recombination and presumed mutant cells. After three Tamoxifen injections (P1–P3), we analyzed the lungs at P7 by scRNA-Seq and histology (Fig. 5A). We clustered and annotated libraries from 9911 cells from wildtype and mutant lungs (Supplementary Fig. 4D and Supplementary Data 7A). The UMAP-plot was consistent with the cell sorting criteria and showed that the inactivation did not affect *Fgfr2* expression in basal and AEC2 cells (Supplementary Fig. 4E–G). All cell-clusters, except cl-2, containing S2 cells, were composed of intermingled wildtype and mutant cells, indicating that *Fgfr2* inactivation in Scgb1a1pos cells does not have off-target effects in other cell-types. The *RFPpos* and *RFPneg* cells of cl-2 showed a conspicuous separation (Supplementary Fig. 4D). The detected *Fgfr2* transcripts in the cells from the *RFPpos* libraries (Supplementary Fig. 4E) suggested escaper cells, which recombined the Rosa26R-Ai14 but failed to deactivate both *Fgfr2* alleles. We confirmed the partial Fgfr2 inactivation by immunofluorescence (Supplementary Fig. 4H), and to reduce noise, we removed the cl-2 and cl-4 cells with *Fgfr2* transcripts from the mutant single-cell library before further analyses.

Clustering equal numbers of secretory cells from wildtype and mutant libraries revealed eight clusters (Fig. 5B, Supplementary Fig. 5A–C, and Supplementary Data 7B, C). These correspond to a single S1-cluster composed of wildtype and mutant cells, two wildtype S2- (WT S2-1 & WT S2-2) and three mutant S2- clusters (KO S2-a, KO S2-b and KO S1/2). In addition, we identified a wildtype and a mutant DP cell cluster (WT DP & KO DP). Differential expression analysis of all wildtype and all mutant airway secretory cells, regardless of clustering, identified 240 statistically significant DEGs, which we categorized based on GO-analysis and previous knowledge (Fig. 5C-D and Supplementary Data 7D–G, 8).

We further related the expression of the affected genes in the mutant airway cells with their expression in the remaining cell types of the whole dataset (Supplementary Fig. 5D). We found a prominent reduction of genes involved in lamellar body and late endosome formation (e.g. *Sftpc, Napsa,* and *Cd74*) accompanied by increased expression of *Hopx,* which is normally expressed in alveolar type-I cells (AEC1)[74], embryonic stalk epithelial progenitors[74,75] and club cells in the intermediate part of the adult mouse trachea[76] (Fig. 5C). Mutant cells also upregulated nuclear genes encoding mitochondrial components of complexes I, III, IV, and V, sodium channel genes (*Commd3, Scnn1a,* and *Scnn1b*[77]) and genes related to autophagy and vesicle trafficking (*Creg1*[78], *Vamp8, Vamp5*[79], *Rab25,* and *Rabac*), which are normally enriched in AEC1s[80] (Fig.5C, Supplementary Fig. 5D and Supplementary Fig. 6A-C). The expression of senescence- and cell-survival-related genes *Cdkn1a* (p21)[81,82], *Bax,* and *Bag1*[83] is also enriched in normal AEC1s and became up-regulated in the mutant airway cells upon Fgfr2-inactivation. Interestingly, *Hopx, Vamp5, Creg1,* and *Scnn1b* transcripts are also detected at low levels in adult wild-type S2 cells, indicating a

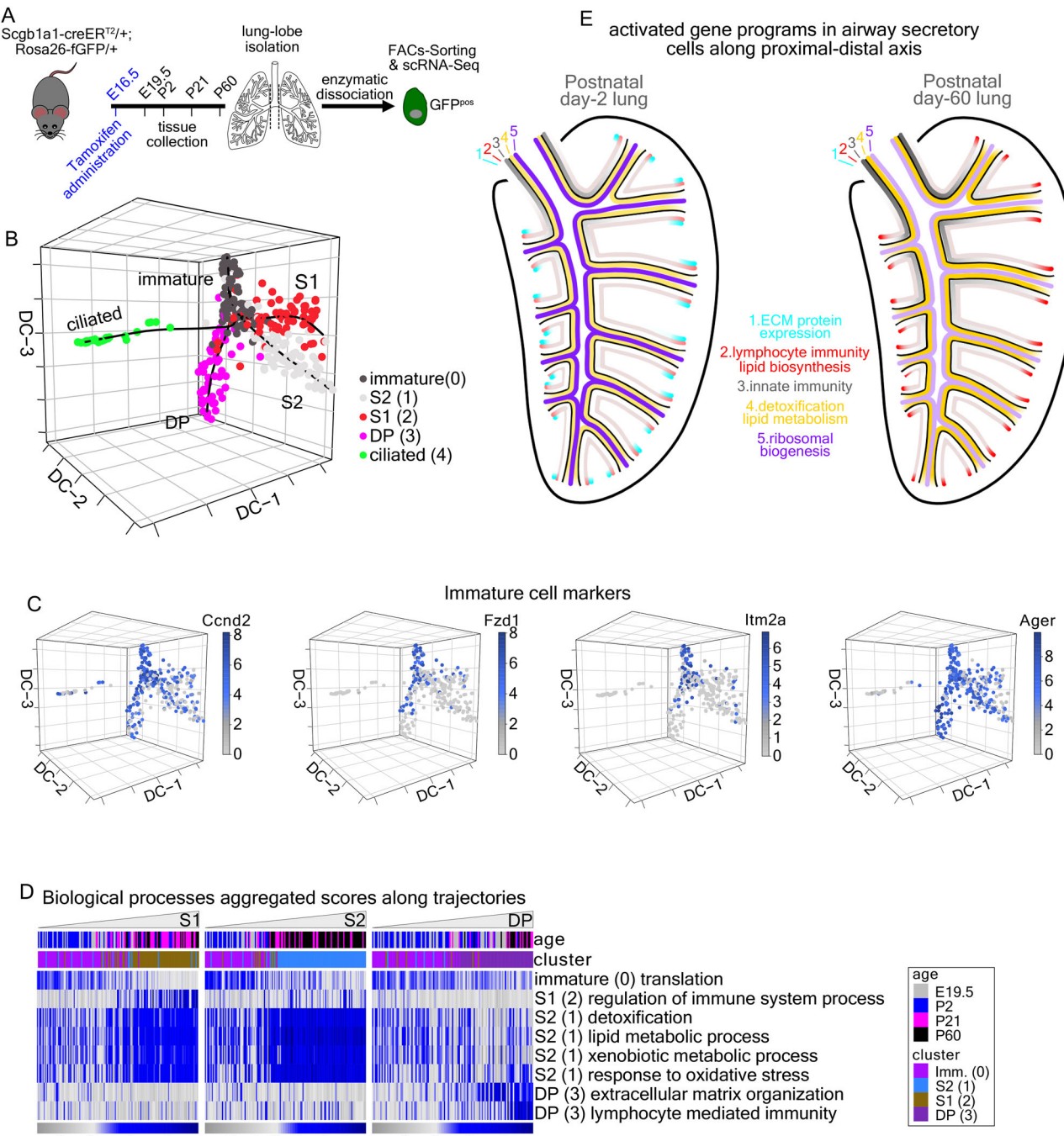

**Fig. 4 | Lineage-tracing of airway secretory cell heterogeneity. A** Experimental outline for the isolation and single-cell RNA sequencing (scRNA-Seq) of the labeled cells from the Scgb1a1-CreER[T2 pos/neg];Rosa26-fGFP[pos/neg] reporter mice. Three embryonic day-19.5, four postnatal-day (P) 2, three P21 and two P60 lungs were analyzed. **B** 3D Diffusion-map plot of 354 full-length, single-cell cDNA libraries. Colors: suggested clusters. Lines: four distinct lineage-trajectories, calculated by Slingshot. **C** 3D Diffusion-map plots of the perinatally expressed genes *Ccnd2*, *Fzd1*, *Itm2a*, and *Ager*. Expression levels: log₂(normalized counts+1) (library size was normalized to 10⁶). Blue: high, Gray: zero. **D** Heatmaps of the aggregated gene expression scores of the indicated biological processes (see Supplementary Data 6). The cells were ordered according to the pseudotime values of the trajectories in (**B**). Blue: high, Gray: low. **E** Synopsis of the gene expression programs activation in airway secretory epithelial cells, along the proximal-distal axis, in the postnatal day-2 (left) and -60 (right) lungs. Color intensity: activation level. Dark: high, Fade: low. ECM: extracellular matrix.

propensity to further activate AEC1 programs (Supplementary Fig. 6D). This suggests that Fgfr2-signaling in perinatal distal airway cells upregulates genes for surfactant biosynthesis, lamellar body and late endosome formation (Fig. 5D). These genes are higher but not exclusively expressed by AEC2s. Fgfr2 also directly or indirectly suppresses genes that relate to mitochondrial function, ion homeostasis, vesicle trafficking and cell-survival, which are normally expressed by AEC1s.

To examine the role of *Fgfr2b* inactivation in the temporal progression of airway cell differentiation, we related the gene expression changes in mutant cells with the expression of these genes in the lineage-tracing experiment (Supplementary Fig. 5E). *Fgfr2b*-inactivation in the airways caused increased expression of ECM protein-encoding genes (*Eln*, *Mgp* and *Mfap4*), which normally show a transient expression along the DP-cell trajectory (Supplementary Fig. 5E) and in

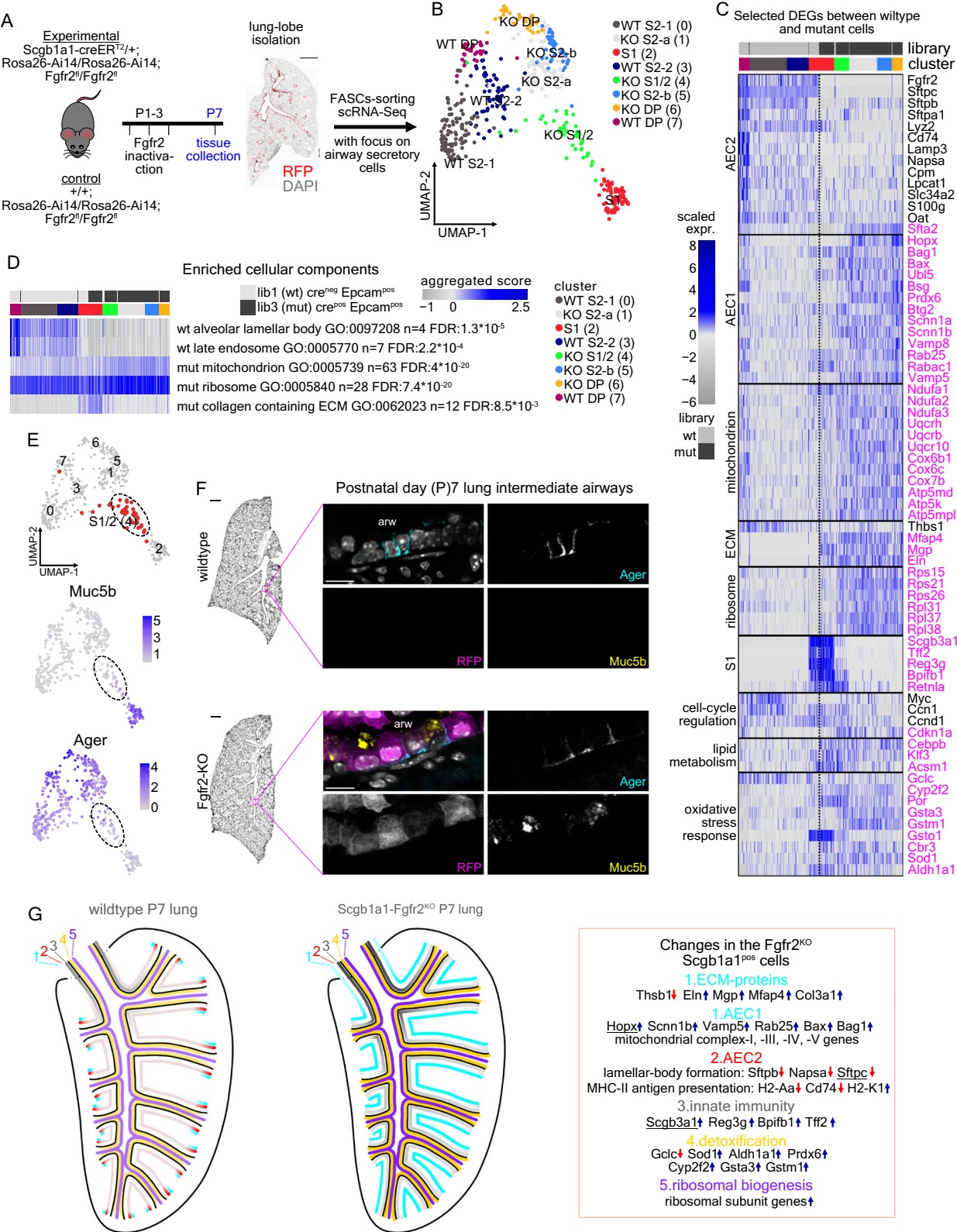

developing AEC1 and AEC2 (Supplementary Fig. 6C) of the normal lung. Similarly, mutant cells failed to downregulate numerous ribosomal-subunit genes, which are highly expressed in all immature lung epithelial cells and gradually decrease as maturation proceeds postnatally (Supplementary Figs. 5E and 6C). These findings suggest that Fgfr2 is required for the normal progression of distal airway epithelial maturation.

An intriguing phenotype upon Fgfr2-inactivation was the appearance of cl-4 (S1/2) that contains predominantly mutant cells (Supplementary Fig. 5B, C) co-expressing moderate levels of innate immunity genes, such as *Tff2, Bpifb, Reg3g, Muc5b*, and *Scgb3a1*. These characteristic S1 markers are normally upregulated in proximal airway cells later in development (Fig. 4). The cl-4 (S1/2) cells also co-expressed *Ager*, whose expression is normally restricted to distal

**Fig. 5 | Fgfr2-inactivation in airway secretory cells causes extended gene expression changes. A** Experimental outline for the perinatal inactivation of Fgfr2 in Scgb1a1$^{pos}$ cells and analysis with single-cell RNA sequencing (scRNA-Seq) and histology. Cells from three wild-type ($n = 3$) and three mutant ($n = 3$) lungs were pooled and used for downstream analyses. **B** UMAP-plot of equal numbers ($n = 229$) of randomly-selected mutant and wildtype airway secretory cells of clusters -2 and -4 in Supplementary Fig. 4F. Colors: suggested clusters. **C** Heatmap showing the expression of selected, differentially expressed genes between the wildtype (library-1) and the mutant (library-3) airway secretory cells. Genes are organized in distinct categories according to previous knowledge and Gene Ontology analysis. Color: scaled expression. blue: high, gray: low. **D** Heatmap of the aggregated scores of selected statistically significant, altered cellular components according to GO-analysis (see Supplementary Data 8). The results are based on the statistically-significant, differentially expressed genes between the wildtype (library-1) and the mutant (library-3) airway secretory cells, using MAST and Bonferroni post hoc test. The cells are ordered according to the clusters (colors as in **B**). Score: blue: high, gray: low. "FDR": false discovery rate by Fisher's Exact test, "*n*": number of genes. **E** UMAP-plots showing the S1/2 cluster-4 (red) and its moderate expression levels for *Muc5b* and *Ager* (dotted region of interest). Expression levels as log$_2$(normalized UMI-counts+1) (library size was normalized to 10.000). **F** (Left) whole lung section overviews of the analyzed postnatal day-7 (P7) lung sections with immuno-fluorescence. The square brackets show the position of the high magnification, confocal microscopy images on the right. Scale-bar: 500 µm. (Right) Single z-step confocal microscopy images of distal airway epithelium from a wildtype and an Fgfr2-mutant lung. Immunofluorescence for Muc5b (yellow) and Ager (cyan). Rosa26-Ai14 (magenta) indicates cells that underwent recombination. Nuclei-DAPI: gray. Scale-bar 5 µm. "arw": airway. Three mutant and two wildtype lungs were analyzed. **G** Schematic representation of the airway epithelium in postnatal day-7 wildtype (left) and Scgb1a1-Fgfr2KO (right) lung, summarizing the observed gene-expression differences. Color intensity: activation level. Dark: high, Fade: low. ECM: extracellular matrix, AEC2: alveolar secretory cell type-2 genes, AEC1: alveolar secretory cell type-1 genes.

airway cells at P7 (Fig. 5E). To confirm the presence of these cells in the mutant lungs and define their topology, we performed immuno-fluorescence analysis for Muc5b and Ager. We detected Muc5b$^{medium}$ Ager$^{medium}$ (S1/2) cells only in the intermediate airways of the mutant lungs (Fig. 5F). This suggests that *Fgfr2*-inactivation in airway cells expands the innate immunity program, altering the patterning of the entire epithelium and not only the gene expression levels of the cells located in distal airways and TBs (Fig. 5G).

## Fgfr2 manipulations in organoids reproduce the in vivo phenotype

To further examine whether Fgfr2b activation is sufficient to induce the anticipated gene expression changes, we established a feeder-free organoid system of airway cells[84]. We isolated RFP$^{pos}$ Pdpn$^{pos}$ and RFP$^{pos}$ Pdpn$^{neg}$ airway cells from the double reporter mice (as in Supplementary Fig.1A), cultured them for twelve days and then applied FGFs or FGFs together with Alofanib, an allosteric Fgfr2 inhibitor[85] for 48 h (Fig. 6A). We analyzed the organoids by immunofluorescence and SCRINSHOT.

The proximal airway-derived RFP$^{pos}$ Pdpn$^{pos}$ fraction produced mainly cystic colonies with thin walls (type-α), which were composed of cells with high *Scgb3a1*, *Muc5b*, and *Bpifb1* but low *Sftpc*, *Ager*, and *Hopx* expression (Fig.6A–D and Supplementary Fig. 7). On the other hand, the RFP$^{pos}$ Pdpn$^{neg}$ cells, from intermediate and distal airways, produced predominantly organoids with thicker walls (type-β) with cells highly expressing *Sftpc*, *Ager*, and *Hopx* and minimal levels of the *Scgb3a1*, *Muc5b*, and *Bpifb1* markers. This suggests that the two fractions of airway secretory cells retain their topology-related characteristics in vitro. This region-of-origin dependent feature of airway cells was also detected in a co-culture assay of the same cell fractions with a lung fibroblast cell line (MLg-2908)[86–88] (Supplementary Fig. 8).

Following this initial characterization, we treated organoids deriving from Pdpn$^{pos}$ or Pdpn$^{neg}$ with an Fgf7/Fgf10 cocktail to induce Fgfr2 activation. In parallel samples, we added the two ligands together with Alofanib to test the reversibility of the ligand effects. We analyzed cells from *Cyp2f2*$^{pos}$, airway-like organoids by SCRINSHOT targeting mRNAs of 9 characteristic airway marker genes. Addition of FGFs for 48 h was sufficient to reduce the innate immunity-related, S1 markers *Scgb3a1*, *Muc5b*, and *Bpifb1*, with more profound changes in the Pdpn$^{pos}$-derived organoids (Fig. 6E and Supplementary Fig. 9A). Alofanib treatment of these cells reversed this effect, suggesting the requirement of FGFR for their suppression. In Pdpn$^{pos}$-derived organoids, the addition of FGFs induced the expression of both *Sftpc* and *Sftpb* (1- and 1.4-fold, respectively, Supplementary Data 9). However, in Pdpn$^{neg}$-derived organoids, which already express highly these two markers, Alofanib reduced their expression, resembling the effect of Fgfr2-inactivation in vivo. Notably, we observed that *Fgfr2* transcripts responded to the activation levels of the pathway, as they were reduced by the addition of FGFs and increased by Alofanib, especially in the Pdpn$^{pos}$-derived organoids. This suggests a feedback loop regulating *Fgfr2* transcription. Unexpectedly, *Scgb1a1* responded differently in the two organoid types, with FGFs reducing its expression in Pdpn$^{pos}$ organoids but slightly inducing it in Pdpn$^{neg}$. Alofanib reversed the effect in both cases. *Cyp2f2* mRNA was reduced by FGF-ligand treatments of both organoid types, but that was reversed only in the Pdpn$^{neg}$-derived organoids, in agreement with the in vivo phenotype in the S2 cells of P7 lungs in vivo (Fig. 5C). Overall, the in vitro treatments suggest that Fgfr2 activation is sufficient to suppress the characteristic S1 marker genes and induce and maintain the expression of S2 markers in organoids resembling the cellular compositions of different airway regions. The differences in the type and the degree in the response of Pdpn$^{pos}$- and Pdpn$^{neg}$- derived organoid cells further argue for intrinsic differences of the organoid cells relating with the topology of the cells they originate from.

## Interplay of FGF and VEGF signaling on airway gene expression

Previous work had shown that Vegfa, through Kdr, suppresses mucus production in intralobar airways postnatally and during regeneration upon naphthalene injury in adult mouse lungs[43]. We found that Fgfr2-inactivation in the perinatal lung reduced *Vegfa* expression in S2 and DP-cells (Supplementary Fig. 9B), suggesting that the effects of Fgfr2-inactivation in the intermediate and distal airways might be mediated by *Vegfa* down-regulation. To address this, we treated the Pdpn$^{neg}$-derived organoids with Vegfa alone or together with the selective Kdr-inhibitor Semaxanib. In parallel cultures, we combined these treatments with the addition of FGFs alone or in conjunction with Alofanib. We analyzed cells within *Cyp2f2*-expressing organoids by SCRINSHOT for the expression of the 9 characteristic airway cell-type markers as above in the FGF treatments. The analysis revealed a complex interplay between VEGF- and FGFR- signaling in marker gene expression regulation. *Scgb3a1* expression was reduced by both VEGF and FGFs individually and was strongest induced by the addition of both inhibitors, suggesting an additive effect of the two RTKs of this S1 marker (Fig. 6F). Vegfa did not affect the expression of the distal *Sftpb* (Fig. 6G) and *Sftpc* markers (Supplementary Fig. 9C), although *Sftpc* expression was slightly reduced by Semaxanib, which might be explained by a known low effect of the inhibitor on Fgfr2 (1250 times less effective than Kdr). The S1 and general airway markers *Muc5b*, *Scgb1a1*, and *Cyp2f2* were reduced by the separate treatments with FGFs or Vegfa, but the combinatorial treatments caused complex responses correlating with similar changes in *Fgfr2* expression in the same cells (Supplementary Fig. 9D). Both Vegfa and FGFs had a minor effect on the expression of *Ager* and *Hopx* (Supplementary Fig. 9E). It was only their combinatorial treatment that caused a statistically significant

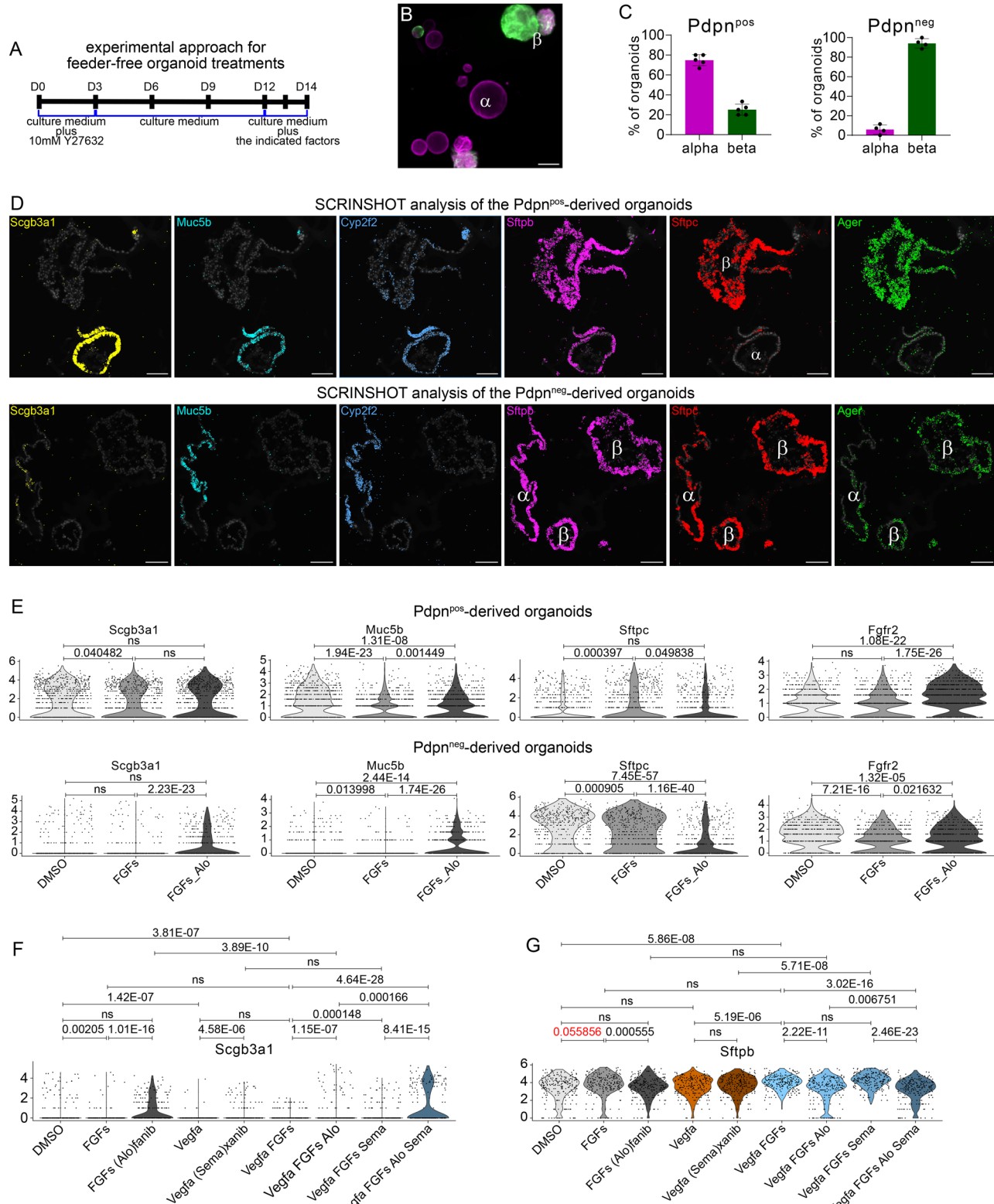

reduction of both markers that became reversed by Alofanib, reproducing the in vivo phenotype in P7 airways. In summary, these results show that the two receptor tyrosine kinases (RTKs) regulate differently the analyzed marker genes, indicating that Vegfa signaling cannot substitute for the Fgfr2-inactivation. The in vitro effect of Vegfa treatment on Fgfr2 expression and the in vivo suppression of *Vegfa* by Fgfr2-inactivation suggest that there is crosstalk between these pathways, in line with the extensive cross-talk of RTK-signaling effectors[89].

This suggests a complicated interaction between VEGF- and FGFR-signaling in the distal airways, where different RTK signaling levels involve cross-regulation of ligands and receptors impinging differently on the regulation of gene-expression.

### Fgfr2 is continuously required in the adult airway epithelium

To explore a homeostatic role of Fgfr2-signaling in the maintenance of graded cell differentiation in the adult airways, we conditionally

**Fig. 6 | Pharmacological FGFR-signaling perturbations in feeder-free organoids reproduce main aspects of the in vivo phenotype. A** Experimental outline for the feeder-free organoid experiments, showing when and which culture media were used. **B** Representative image of organoids from Scgb1a1creER-Ai14^pos Pdpn^neg cells, showing type-α (α) and a type-β (β) organoids. RFP (Scgb1a1creER-Ai14): magenta, GFP (Sftpc): green. Scale-bar: 500 μm. **C** Bar-plot of the morphometric analysis of organoid types according to the sorting criteria (Ai14^pos Pdpn^pos cells: 5 replicates, 26 organoids; Ai14^pos Pdpn^neg cells: 5 replicates, 149 organoids). Error bars: standard deviation from the mean (Mean ± SD). **D** SCRINSHOT analysis of Pdpn^pos (top) and Pdpn^neg (bottom) organoids showing the detected mRNAs of the indicated genes. Scgb3a1: yellow, Muc5b: cyan, Cyp2f2: light-blue, Sftpb: magenta, Sftpc: red, Ager:

green, nuclei-DAPI: gray. Scale-bar 100 μm. **E** Violin-plots showing the detected mRNAs for the indicated markers in Pdpn^pos (top) and Pdpn^neg (bottom) organoids. For Pdpn^pos $n$ = 770 cells/condition and for Pdpn^neg $n$ = 515 cells/condition. Expression: log$_2$(SCRINSHOT dots+1). Statistical pairwise analyses with MAST. The numbers over the plots show the Bonferroni adjusted $p$-values, calculated by MAST. "ns": non-significant (>0.05). **F, G** As in (**E**) for the expression of Scgb1a1 and Sftpb in Pdpn^neg organoids. $n$ = 245 cells/condition. In (**G**), Bonferroni adjusted $p$-value between DMSO and FGFs (0.055856) indicates a tendency for increased expression. The results of all statistical analyses and exact $p$ values are provided in Supplementary Data 9. Source data are provided as a Source Data file.

deactivated *Fgfr2* in P60 mice. *Scgb1a*- creER^T2-mediated recombination resulted in RFP-expression and loss of Fgfr2 staining in both proximal and distal airway compartments (Fig. 7A and Supplementary Fig. 10A). We used a representative panel of 9 marker genes, including *Fgfr2*, to detect their expression in situ. The panel included *Scgb3a1* and *Muc5b* for S1 cells, *Scgb1a1*, *Cyp2f2* for S1 and S2 cells, *Sftpb* and *Sftpc* for S2 and AEC2, and the AEC1 markers (*Hopx and Ager*). Equal numbers of control and mutant airway cells were grouped into five clusters that correspond to S1 (cl-1), S2 (cl-0 and -2), distal airway *Ager*^pos cells (cl-3) and to DP cells (cl-4) (Fig. 7B). Each cell in the UMAP-plot was also annotated according to its position as proximal, intermediate or distal (including TBs) and its condition as *Fgfr2* mutant or control (Fig. 7C, D). As in the P7 lungs, S2 mutant cells downregulated *Scgb1a1* and *Sftpb* and induced *Hopx* expression (Fig. 7E, F and Supplementary Data 10), driving S2-cell separation into the cl-0 and -2, which were localized in intermediate and distal airways of wildtype and mutant lungs, respectively (Fig. 7G, H). The reduced *Scgb1a1* and *Sftpb* expression upon Fgfr2-inactivation was also confirmed at the protein level by immunofluorescence (Supplementary Fig. 10B). The cl-1 group contained S1-cells from both wildtype and mutant lungs, which were localized predominantly in the most proximal parts of the airway network (Fig. 7G, H). We observed a minor but insignificant tendency for more S1 cells in the intermediate airways of the mutant lungs and a slight increase of *Muc5b*^pos but not *Scgb3a1*^pos cells (Supplementary Fig. 10C, D). This suggests that, in contrast to the P7 lungs, Fgfr2b is no longer required to restrict the innate immunity gene-expression program in adult lungs. In distal airway domains, there was a statistically significant reduction of DP-cells (cl-4) upon Fgfr2-inactivation (Fig. 7H). This could result from the down-regulation of *Sftpc* or by cell apoptosis upon Fgfr2-inactivation, as previously reported[34].

In summary, these results indicate that Fgfr2 is required for the continuous expression of *Scgb1a1* and *Sftpb* in the intralobar airway epithelium in addition to the maintenance of DP-cells. The minor induction of the S1 markers and the unaffected expression of *Ager* suggest that *Fgfr2b* is dispensable for maintaining the innate immunity patterning and maturation state of the already mature adult airways.

## Discussion

ScRNA-Seq and spatial gene expression analyses of lung secretory cells revealed distinct programs of co-expressed genes underlying the functional characteristics of the tissue. The quantitative spatial analysis of characteristic markers for these programs revealed at least two opposing and partially overlapping gradients of hundreds of genes on top of a uniformly activated detoxication program. This creates a continuum of cell-states along the proximal-distal axis of the organ. The genes are involved in innate immunity, cytokine production and response to cytokines in proximal regions and antigen presentation, lipid synthesis and surfactant production in the distal ones. Similarly, recent reports showed differential expression patterns of proximal and distal markers along the human airway epithelium[15,16,90,91]. However, because of the size of the human lung and the lack of stereotyped branching landmarks in the human samples, the spatial analysis of

three different locations along the proximal-distal axis of the human lung are suggestive but do not prove the presence of opposing tissue-scale gradients along the airway network[16].

Why may these gene expression gradients be relevant for airway structure and function? Firstly, the slope correlates with the tapering of airway branches, suggesting that graded gene expression programs may control branch size and shape, facilitating seamless airflow to the alveolar compartments. Second, the differentially localized expression of different types of immunity programs suggests that proximal cells are better endowed to present immediate innate responses and cytokine signaling. In contrast, the distal ones are more specialized for antigen presentation. The compartmentalization of immunity-related functions correlates with the higher expression of mucin coding genes and greater abundance of multiciliated cells in proximal regions, where pathogens become trapped, targeted by antimicrobial peptides and propelled out of the airways. Escaping pathogens may be further detected by distal airway cells, internalized and presented to lymphocytes, activating slower but long-lasting immune responses. Third, the gradients may reflect developmentally controlled positioning of cells expressing moderate levels of specification genes in intermediate positions of each branch in the airway network. Such cells may efficiently and rapidly de-differentiate to repair local damage caused by pathogens and inhaled toxic substances, replenishing the epithelium with progeny featuring the same topology-related characteristics. Our in vitro culture experiments support this notion, but the present analysis has not identified unique marker genes for efficient labeling and isolation of these cells.

The lineage tracing of embryonic secretory progenitors combined with scRNA-Seq indicates that airway gene-expression programs are established sequentially, suggesting that they are hierarchically coupled and regulated. Immature airway secretory cells first upregulate a detoxification-related genetic program, which is retained and increased along the airways but not in DP-cells at the bronchoalveolar junctions. Innate immunity-, lipid metabolism- and lymphocyte-mediated-immunity gene programs become selectively established later in proximal or distal cells. Conversely, earlier developmental programs of genes encoding ribosomal proteins or ECM proteins become downregulated as the maturing secretory cells reach the end of their differentiation trajectory. The expression of these genes is retained longer in DP-cells, where it becomes downregulated later. The identification of specific transcription factor encoding genes that are activated or downregulated along the developmental trajectories leading to different secretory cell states provides an entry point towards the functional dissection of airway cell differentiation.

The conditional Fgfr2-inactivation in the postnatal airway secretory epithelium suggests a central role for Fgfr2-signaling in the early postnatal, distal airway cell differentiation and patterning. First, the graded distal programs encoding surfactant production and endosomal vesicle traffic, which are normally active in secretory cells of intermediate and distal bronchioles and alveoli, become severely reduced. Instead, the cells in mutant airways upregulate genes encoding mitochondrial proteins and autophagy that are normally

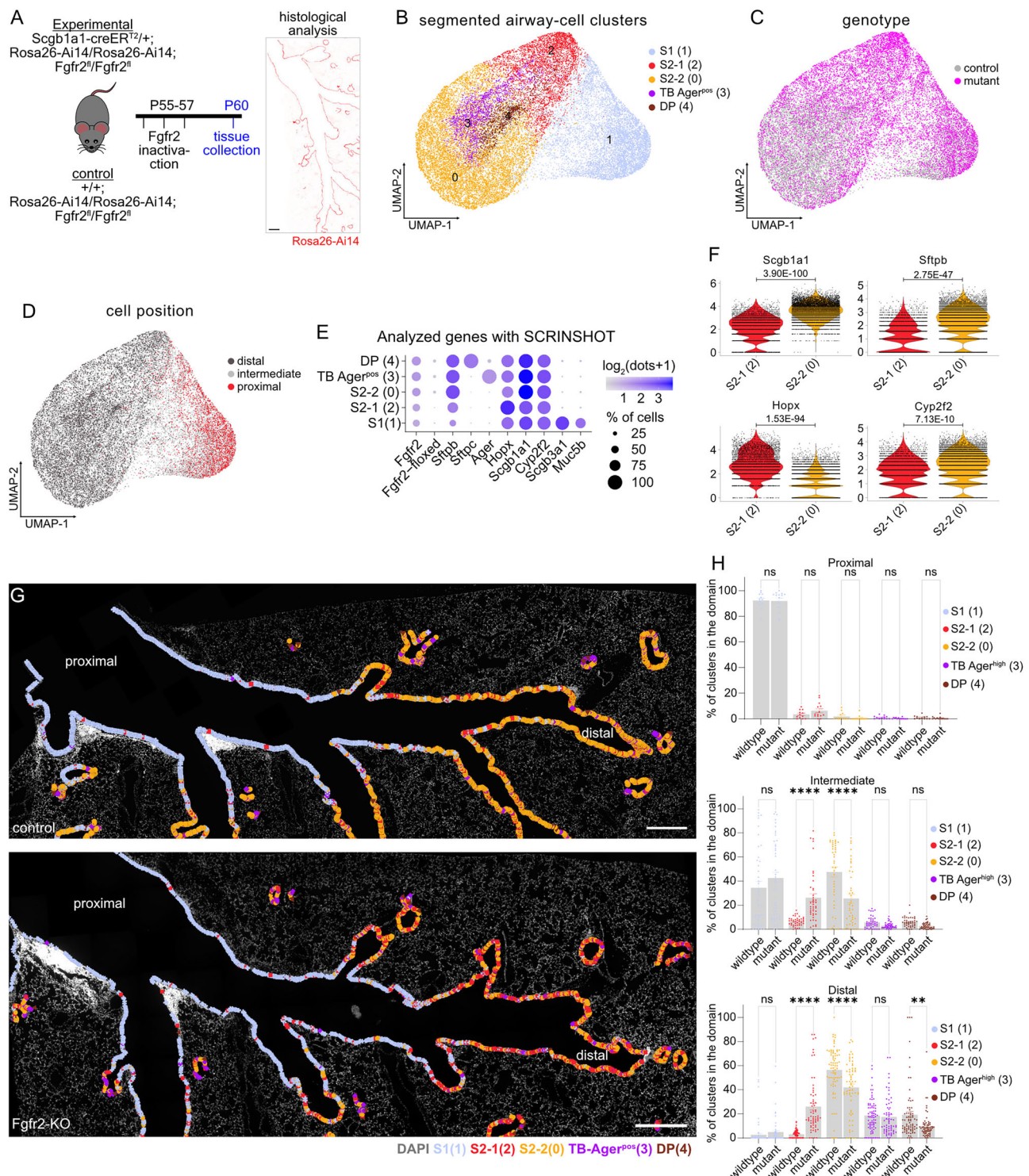

enriched in AEC1s. This phenotype is similar to the one described for the perinatal Fgfr2-inactivation in AEC2s, which reprograms them to AEC1s[34]. Second, our analysis in the mutant airways reveals a previously unappreciated role of Fgfr2-signaling in all S2-cells, because the timed repression of ribosomal and ECM gene expression remains active in all mutant S2-cells. This suggests that Fgfr2-signaling promotes the progression of airway secretory cell differentiation. The reduced expression of *Myc*, *Atf4*[92], and *Ets1* in mutant airway cells may be linked to lower *Slc7a11* expression, affecting the detoxification ability of the cells by compromising their ability to exchange intracellular glutamate with extracellular cystine for glutathione synthesis[93]. Both *Slc7a11* and *Gclc*,

a rate-limiting enzyme of the first step of glutathione biosynthesis[94] were reduced upon *Fgfr2*-inactivation. The reduced glutathione levels might, in turn, indirectly up-regulate other detoxification- and oxidative stress-related genes, like *Sod1*, *Aldha1*, *Gsta3*, and *Gstm1*, in a presumed compensatory mechanism to spare the mutant cells from reactive oxygen species and lipid peroxidation. Further functional studies are needed to understand the role of Fgfr2 in the regulatory mechanisms behind the detoxification machinery in airway epithelium. Finally, mutant cells in the intermediate domains of the airway network moderately activate innate immunity genes, causing an expansion of that gradient and resulting in supernumerary S1/2 cells.

**Fig. 7 | Fgfr2-inactivaiton in the adult airway epithelium. A** Experimental outline of the Fgfr2 inactivation in adult Scgb1a1$^{pos}$ cells in addition to a representative image of the uniform RFP (Rosa-Ai14) expression along the intralobar airway network, confirming the recombination specificity predominantly in the airways. UMAP-plots of equal numbers of airway cells, from three wildtype and three Fgfr2-KO lungs (13900 cells/condition) analyzed by SCRINSHOT. **B**: clusters, (**C**): experimental condition, (**D**): cell position in proximal, intermediate or distal (including terminal bronchioles (TBs)) airways. **E** Balloon-plot of the analyzed genes in the annotated cell clusters. "Fgfr2-floxed": signal from a padlock probe that detects the deleted exon in mutant cells. Balloon size: percent of positive cells. Color intensity: log$_2$(SCRINSHOT dots+1) (blue: high and gray: low). **F** Violin-plots showing the detected mRNAs for the indicated markers in the two S2 clusters. Expression: log$_2$(SCRINSHOT dots+1). Statistical pairwise analyses with MAST. The numbers over the plots show the Bonferroni adjusted p-values. **G** Images showing the spatial distribution of the cell-clusters along the airway network. Colors as in (**B**). Nuclei-DAPI: gray. **H** Bar-plots showing the percent of the clusters within each region. 2-way ANOVA statistical analysis with Šídák's multiple comparisons test. Error bars: standard error from the mean (Mean ± SEM). Bonferroni adjusted p-value: "ns" non-significant (>0.05), ** $p < 0.01$, **** $p < 0.0001$. Proximal $n = 12$ wildtype and $n = 14$ mutant regions. Intermediate $n = 42$ wildtype and $n = 43$ mutant regions. Distal $n = 76$ wildtype and $n = 66$ mutant regions. Results from three independently analyzed lungs ($n = 3$) per condition. The exact p-values with the order appearing in the figure are "Proximal": 0.9998, 0.3346, 0.9245, 0.9959, 0.9995; "Intermediate": 0.2608, <0.0001, <0.0001, 0.9769, 0.9561; "Distal": 0.9173, <0.0001, <0.0001, 0.9970, 0.0022. Scale bars: 500 μm. Source data are provided as a Source Data file.

In agreement with the in vivo phenotype, activation of the FGFR-signaling in feeder-free organoids was sufficient to suppress innate immunity genes and up-regulate surfactant protein genes. However, these organoids also suggest cell-intrinsic properties of airway cells to respond differently to extracellular signals. Fgfr2-stimulation of organoids deriving from proximal airway cells is insufficient to induce high expression levels of surfactant encoding genes, similar to those in Pdpn$^{neg}$ cells. This correlates with the low expression levels of the receptor in proximal Pdpn$^{pos}$ cells. These intrinsic differences in responses to extracellular signals correlate with the spatial position of the cells on the airway axis and suggest the presence of an under-explored regional specification program that establishes different gene expression subdomains among the embryonic airway cells marked by Sox2. We speculate that the graded expression of differentiation programs in the airways is specified by the interplay of underexplored cell-intrinsic regional programs with the external signals deriving by surrounding cells. Such an ancient patterning scheme involving intrinsic regional factors and extrinsically stimulated RTK signaling operates in the branched airway network of Drosophila[95]. Given that FGFs are secreted from distal stromal cells and FGFR2b is localized in adjacent distal epithelial cells, the repression of innate immunity genes in intermediate regions may involve relaying signaling mechanisms, where FGFR activates the expression of other signaling molecules. These may include Vegfa that signals through Kdr to suppress mucus cell fate during the early postnatal life and adult airway regeneration[43]. We tested the potential relationships of VEGFA and FGF in vitro in the feeder-free organoids. Like FGFs, Vegfa could down-regulate some of the innate immunity genes in vitro, but FGFR-signaling had a unique role in the regulation of the remaining gene-expression programs in the airway epithelium, since the Fgfr2-inactivation was only partially reverted by Vegfa supplementation. In vivo Fgfr2 is required for the upregulation of Vegfa expression, and in vitro activation of both FGFR- and KDR- signaling had a negative impact on Fgfr2 expression, indicating complex crosstalk relationships between different RTK-pathways[89].

The Fgfr2-inactivation in the adult airway epithelium indicated that it is also required for the continuous maintenance of the distal secretory program after the establishment of the postnatal gene expression patterns. The upregulation of *Hopx* in the adult *Fgfr2*-mutant lungs may also be connected to the regulation of surfactant protein genes, as it was shown that Hopx directly suppresses their transcription[96].

Recent work on the temporal and spatial organizations of epithelial heterogeneity at the single cell level begins to reveal cellular and molecular gradients along the anatomical axes of several endodermal epithelial tissues[97,98]. The pathogenesis of chronic obstructive pulmonary disease involves the proximalization of distal respiratory bronchioles, disrupting the ordered organization of the airway epithelium[99]. Our spatial and temporal analysis of secretory cell heterogeneity and its developmental control in the mouse offers a framework for studying epithelial organization in whole tissue sections to better understand cellular roles in tissue-level pathology.

## Methods

### Animal models and tamoxifen administration

All mouse experiments were performed according to Swedish animal welfare legislation and German federal ethical guidelines. The Northern Stockholm Animal Ethics Committee approved the project (Ethical Permit numbers N254/2014 and 15196-2018). The Research Animal Ethics Committee in Gothenburg approved the analyses of germ-free (GF) mice (Ethical Permit number 4805-23). The GF mice were maintained in flexible film isolators (Class Biologically Clean, Madison, WI, USA). GF status was monitored regularly by aerobic and anaerobic culturing and PCR for bacterial 16S rRNA. All mice were group-housed in a controlled environment (under constant humidity (50-60%) and temperature (21 ± 2 °C) and with a constant (year-around) 12-h light/dark cycle), with free access to autoclaved chow diet and water. Breeding and experiments performed in JLU, Giessen, Germany, were under the Ethical Permit with number GI 20/10, Nr. G 21/2017. For the lineage-tracing experiments, we used Scgb1a1-CreER$^{T2\ pos/neg}$;Rosa26-fGFP$^{pos/neg}$[22] mice. Noon of the day of the vaginal plug was considered as embryonic day (E) 0.5. We induced recombination by one oral dose (gavage) of Tamoxifen solution in corn oil (30 mg/kg body weight) on E16.5, as described previously[22]. For the analysis of adult-lung epithelial heterogeneity and organoid cultures, we used Scgb1a1-CreER$^{het}$; Rosa26-Ai14$^{het}$; Sftpc-fGFP$^{het}$ adult mice[39,40] and administered one Tamoxifen dose (100 mg/kg body weight), 72 h prior to tissue collection. Experiments for Fgfr2-inactivation were performed using Scgb1a1-CreER$^{T2\ pos/neg}$;Rosa26-Ai14$^{pos/pos}$;Fgfr2b$^{fl/fl}$ and Scgb1a1-CreER$^{T2\ neg/neg}$; Rosa26-Ai14$^{pos/pos}$; Fgfr2b$^{fl/fl}$ mice. Tamoxifen was daily injected, subcutaneously (87 mg/ kg body weight) on P1-3 and intraperitoneally for P55-57. All the experiments have been performed using C57Bl/6J mice of both sexes. Wildtype mice were obtained by Charles River Laboratories, Scgb1a1-CreER$^{T2}$ and Rosa26-fGFP transgenic lines were kindly provided by Dr. Emma Rawlins and Fgfr2b$^{fl/fl}$ line by Prof. Saverio Bellusci. The Rosa26-Ai14 mouse line was purchased from The Jackson Laboratory (007914).

### Tissue collection

Animals were euthanized by an intraperitoneal injection of anesthesia overdose, followed by incision of the abdominal vein. For embryonic lungs, we did not perform heart perfusion. For postnatal lungs, the chest was opened, and the left atrium was excised. Lungs were perfused through the right ventricle of the heart with ice-cold PBS 1X pH7.4, using a 26 G needle and 5 ml syringe until they became white. Lungs were inflated with a mixture of 4% PFA:OCT (2:1 v/v) using a 20 G catheter (Braun, 4251130-01), until the accessory lobe was expanded. The trachea was ligated (with silk 5/0 Vömel thread, 14739) and tissues were later fixed. For histological analysis, tissues were collected on E19.5, on post-natal day 2 (P2), 5 (P5), 7 (P7), 21 (P21), and 60 (P60).

Embryonic and P2 tissues were fixed with freshly prepared 4% Paraformaldehyde (Merck, 104005) solution in PBS 1X pH7.4 (Ambion, AM9625) for 4 h. Later stages were fixed for 8 h. Thereafter, the tissues were placed in OCT: 30% sucrose in PBS (2:1 v/v) over-night (O/N) at 4 °C with gentle shaking and frozen in OCT (Leica Surgipath, FSC22), using plastic molds (Leica Surgipath, 3803025), by placing them in isopentane and dry ice. Tissue-OCT blocks were kept at −80 °C until sectioning.

## Tissue dissociation and cell isolation

For full-length (Smart-Seq2) library preparation[67], the left lungs were used for enzymatic digestion, and the right lungs were treated as described above for histological analysis. For cell culture and droplet-based scRNA-Seq, both lungs were processed for digestion. Briefly, we cut the lungs into small pieces using a razor blade and digested them with elastase (Worthington, LS002292) and DNase-I 0.5 mg/ml (Sigma-Aldrich, DN25) in HBSS (Gibco, 14175), at 37 °C for 1 h with rotation. An equal volume of HBSS++ [HBSS (Gibco, 14175), supplemented with 2% 0.2 μm filtered FCS (Gibco, 10500), 0.1 M HEPES (Sigma-Aldrich, H0887), antibiotics (Gibco, 15240096) and EGTA 2 mM was added, and the suspension was mixed gently. Then, the cells were centrifuged at 800 ×g for 10 min at 4 °C. The supernatant was removed with a serological pipette, and cells were resuspended in HBSS++. Viability was tested using trypan blue (Sigma-Aldrich, T8154) (1:1 dilution), and the presence of fluorescence-positive cells was evaluated using a fluorescence microscope. Before sorting, cells were passed through a 100 μm BD Falcon (BD Biosciences, 340610) to remove cell aggregates.

For cell-sorting, we used a BD FACSARIA III cell-sorter with 100 μm nozzle using single-cell sorting purity. Cells from all stages were isolated according to GFP and/or Tomato (for Rosa-Ai14 mice) expression. Non-transgenic and single-transgene (either Scgb1a1-CreER; Rosa-Ai14 or Sftpc-GFP) positive animals were used for instrument calibration. For cell culture experiments, cells were sorted in HBSS++ medium and for droplet-based sequencing, we omitted EGTA and HEPES, according to 10× Genomics instructions.

## scRNA-Seq of adult lung cells

Droplet-based scRNA-Seq was carried out with Chromium Next GEN Single Cell 3′ Kit version 3 (10x Genomics), at the Eukaryotic Single Cell Genomics Facility at SciLifeLab, Sweden. The samples were processed with cellranger-4.0.0 pipeline (10x Genomics). The reads were mapped to a custom mouse (GRCm38) reference genome that contained GFP and Ai14 cassette (RFP and WPRE sequences) sequences. The reference genome was created with the 10× Genomics "cellranger mkref", and the mapping of the reads was done with the "cellranger count" function using default settings.

## scRNA-Seq analysis of adult lung cells

For the analysis of the droplet based scRNA-Seq dataset, we initially applied filtering criteria to filter out low quality cells and contaminants (Sftpc-GFP[pos] library: GFP-UMIs > 4, number of detected genes > 2500 and <5500 and percent of mitochondrial genes >0 and <7.5; Scgb1a1-CreER:Rosa-Ai14[pos] Pdpn[neg] library: RFP-UMIs > 4, number of detected genes > 2500 and <5500 and percent of mitochondrial genes >0 and <7.5; Scgb1a1-CreER:Rosa-Ai14[pos] Sftpc-GFP[pos] library: RFP-UMIs > 4 and GFP-UMIs > 4, number of detected genes > 2500 and <5500 and percent of mitochondrial genes >0 and <7.5; Scgb1a1-CreER:Rosa-Ai14[pos] Pdpn[pos] library: RFP-UMIs > 4, number of detected genes > 3000 and <5500 and percent of mitochondrial genes >0 and <5). Genes with less than 50 counts in all cells were removed, and the counts were transformed using the SCTransform[100] function in Seurat[101], with 4000 variable genes and regressing out the number of counts, detected genes and the percent of mitochondrial counts. The first 50 principal components were used for dimension reduction and clustering, setting the number of neighbors to 25 and the resolution to 0.2. MAST[102]

was used to identify DEGs after library normalization to 10.000 and log$_2$-transformation.

We used an equal number of cells/cluster for the trajectory analysis and ran diffusion maps with Destiny[45], implemented with scMEGA[103]. We used the first 16 principal components and $k = 25$. We used the first three diffusion-map components for visualization and downstream analyses. We calculated the principal curves ("getCurves" function), the pseudotime estimates ("slingPseudotime" function) and the lineage assignment weights ("slingCurveWeights" function) with Slingshot[104]. We identified differentially expressed genes with the "fitGAM" function of tradeSeq[105]. For multiple trajectories, we used the "patternTest" and for one the "associationTest" functions. The genes were ordered based on the hierarchical clustering ward.D2 method, using "hclust" function in fastcluster package[106] and plotted using a custom script. The "clusterboot" function of fpc package[107] was used to calculate the stability values of gene-modules. GO-analyses were done at http://geneontology.org/ selecting as organism the Mus musculus and using default settings. The Fisher's Exact test calculates the False Discovery Rate (FDR). Aggregated gene expression scores of genes in modules and biological processes were calculated with the "AddModuleScore" function in Seurat. For Balloon-plots and heatmaps, we used the "DotPlot" and "DoHeatmap" functions in Seurat, in addition to the pheatmap-package[108].

## Full length scRNA-Seq

Single-cell library preparation was done according to Smart-Seq2 protocol[67,109] with some modifications. Cells were sorted in 96-well plates (Piko PCR Plates 24-well, Thermo Scientific, SPL0240 and Plate Frame for 24-well PikoPCR Plates, Thermo Scientific, SFR0241). Each well contained Triton-X100 (0.2%) (Sigma-Aldrich, T9284-100ML), ERCC RNA Spike-In Mix (1:400.000) (Life-Technologies, 4456740), Oligo-dT30 VN (1.25 μM) AAGCAGTGGTATCAACGCAGAGTAC(30 x T) VN, dNTPs (2.5 mM/each) (Thermo Scientific-Fermentas, R0192) and Rnase Inhibitor (1U/μl) (Clontech, 2313 A) in 4 μl final volume. After sorting, strips were covered with Axygen PCR-tube caps (VWR, PCR-02-FCP-C), centrifuged and placed on dry ice until storage at −80 °C for further use. For cell culture, cells were sorted into HBSS++ buffer and kept on ice until they were processed for culture. To optimize Smart-Seq2[67] for mouse primary lung cells that are small and contain a small amount of RNA, we used 50% less Oligo-dT30 VN and the cDNA synthesis was divided into two steps, the first was without TSO LNA, and the second contained 1 μM TSO LNA and an additional 40U of SuperScript II RT (Thermo-Fisher Scientific, 18064071). The reaction lasted 30 min at 42 °C. Then, the enzyme was deactivated at 70 °C for 15 min. For Pre-Amplification PCR, we used the KAPA HiFi Hotstart ReadyMix (2x) (KAPA Biosystems, KK2602) and the ISPCR-primer AAGCAGTGGTATCAACGCAGAGT. PCR included 21 cycles, and the total volume increased to 50 μl in order to reduce the concentration of the unused Oligo-dT30 VN and TSO-LNA primers.

Tagmentation and indexed library amplification were done with Nextera® XT DNA Library Preparation Kit (Illumina, FC-131-1096) and Nextera® XT Index Kit (96 indexes, 384 samples) (Illumina, FC-131-1002) according to the manufacturer's protocol (with 2.5 x volume reduction in all reactions). For tagmentation, we used 50 pg of the libraries, as it was indicated by the 500–9000 bp fraction of the library (Bioanalyzer).

Sequencing was done with Illumina 2500 HiSeq Rapid mode using paired-end (2 × 125 bp) and single-end (1 × 50 bp) reading. For downstream analyses, we used one strand of paired-end libraries and trimmed the reads to 50 bp.

## scRNA-Seq analysis of Smart-seq2 dataset

We initially kept the libraries with >40% uniquely mapped reads to a reference genome that contained GFP and ERCC sequences and removed *Esr1*, as an artifact, because of sequence similarities with

Scgb1a1-CreER$^{T2}$ transgene and *Xist*. Individual sequencing datasets were filtered regarding the number of detected genes (lower threshold: 2000 genes and upper threshold 10000 (P2272, P2661) and 6000 (P3504, P7657)). Then, we filtered out the libraries with more than 200 counts of Pecam1 as not epithelial contaminants. Finally, we removed libraries with more than 7.5% of mitochondrial gene counts, resulting in 354 libraries for downstream analysis.

We used SCT-transformation in Seurat with 3000 variable genes and regressed out the number of counts and detected genes and the percent of mitochondrial counts. The first 20 principal components were used for dimension reduction, setting the number of neighbors to 12 and resolution to 1. Diffusion maps were produced as in the adult dataset using the first 12 principal components and $k = 12$. For the identification of DEGs, we used the MAST analysis in Seurat. For the trajectory analyses, we used Slingshot, setting as root the cluster-0 (embryonic) and end-point clusters the -2 (S1), -1 (S2), -4 (ciliated), and -3 (DP). The diffusion-map 3D-plots were created with scatter3D function of scatterplot3d[110]. GO-analyses and aggregated scores were produced as in the adult dataset.

### scRNA-Seq of *Fgfr2*-inactivated airway epithelial cells

We followed the procedure for tissue dissociation and cell isolation as in the other FACs-sorting experiments. Single-cell suspensions from three P7 Scgb1a1-CreER$^{T2 neg/neg}$; Rosa26-Ai14$^{pos/pos}$; Fgfr2$^{fl/fl}$ mice were pooled and used as negative control samples. Three P7 and Scgb1a1-CreER$^{T2 pos/neg}$; Rosa26-Ai14$^{pos/pos}$; Fgfr2$^{fl/fl}$ mice were combined and used as experimental groups. Cells were counted with a Biorad cell counter, blocked with TruStain FcX™ PLUS (Biolegend, 156604), stained with a PE/Cyanine7 anti-mouse CD326 antibody, and washed according to the manufacturer's protocol (Biolegend, 118216). The negative control samples were sorted based on Epcam positivity, and from the experimental groups, we isolated Epcam$^{pos}$-Ai14$^{neg}$ and Epcam$^{pos}$-Ai14$^{pos}$ cells. The isolated cells were processed with the Chromium Next GEM Single Cell 3′ Reagent Kits v3.1 (10xGenomics), following the manufacturer's instructions and targeting 7000 cells/well. The produced libraries were sequenced with a NovaSeq 6000.

### scRNA-Seq analysis of the *Fgfr2*-inactivated epithelial cells

We initially processed all cells and filtered out genes that were expressed in fewer than 5 cells and followed the same analysis approach as in the lineage-tracing dataset, using 4000 variable genes. We removed Krt13$^{high}$ esophageal/tracheal basal cells, Ptprc$^{pos}$ immune and Col1a2$^{pos}$ mesenchymal cells as contaminants. Filtered cells were re-clustered after filtering out genes that are expressed in less than 20 cells and selecting the 5000 most variable genes. We used the first 20 principal components, $k = 15$ and resolution = 0.2. Then, we selected the airway secretory clusters for downstream analyses as described for the other datasets, using 600 variable genes, 15 principal components, resolution = 0.99 and k = 8.

### scRNA-Seq analysis of the GSE149563

For the analysis of the publicly available GSE149563 scRNA-Seq dataset, we analyzed each timepoint individually with Seurat, using 4000 variable genes and 50 top principal components. We used DoubletFinder[111] package in R to identify and remove multiplets. The postnatal datasets were integrated and processed for clustering and differential expression analysis as in the other datasets. The epithelial clusters were further filtered to remove possible endothelial (Pecam1$^{pos}$) and mesenchymal (Col1a2$^{pos}$) cells and re-clustered, selecting the 4000 most variable genes. We used the 50 first principal components, $k = 25$ and resolution = 0.6.

### Organoid cultures

Lung digestion and cell sorting were performed as above, including the Dead cell stain NucRed (Thermo, R37113) to sort out dead cells. Sorted epithelial cells (100–600 cells/well) were mixed with the Mlg2908 (ATCC, CCL-206) mouse lung fibroblasts (10$^4$ cells/well), as described before[112]. Colonies were then fixed in 4% PFA O/N and placed in 30% sucrose solution for 24 h. Freeze-thawing and gentle pipetting were performed twice to remove the Matrigel. Colonies were then incubated with 30% sucrose and 30% OCT overnight and embedded in OCT. The blocks were sectioned at 12–14 μm for immunofluorescence.

For the feeder-free organoid experiments, we isolated cells from the double reporter mice with a BD FACSMelody™ Cell Sorter. Cells were centrifuged for 10 min at 800 × g 4 °C and resuspended in culture medium. Equal volume of Matrigel (Corning 354230) was added to the cell suspension with gentle mixing. We transferred 25 μl of the mixture at the bottom of individual wells of 48-well plates (Costar 734-1607) and placed them in the cell culture incubator (37 °C, 5% CO$_2$) for 20 min to allow Matrigel solidification. Then 200 μl of culture medium was added containing DMEM/F12 (Thermo 11039021), 1x B27 (Thermo A3582801), 5% FBS (Thermo 10500064), 15 mM HEPES (Thermo 15630056), 0.03% NaHCO3 (Sigma Aldrich S8761), 1x P/S/Amph (Thermo 15240062), 30 ng/ml HGF (R&D Systems 2207-HG-025), 100 ng/ml Noggin (R&D Systems 6997-NG-025 and StemCell Technologies 78061), 10 μM SB431542 (Sigma Aldrich 616461), 3 μM CHIR99021 (Sigma Aldrich SML1046-5MG) and 10 μM Y27632 (R&D Systems 1254). The medium was changed every 2 days until day-12, omitting the Y27632. For the treatments we used 50 ng/ml FGF10 (R&D Systems 6224-FG-025) and 50 ng/ml Fgf7 (R&D Systems 5028-KG-025), 100 ng/ml Vegfa (StemCell Technologies 78102), 200 nM Alofanib (Selleck Chemicals S8754), 1.3 μM Semaxanib (MedChemExpress HY-10374) and DMSO (Sigma-Aldrich D2650). We treated the organoids every 24 h for 2 days (until day-14) and then fixed them with 4% PFA overnight (O/N) at 4 °C. We washed the cultures twice with PBS 1X and imaged them with a Zeiss Axio Observer Z1 inverted microscope, equipped with a 5x EC Plan-Neofluar lens, an automated stage and a Hamamatsu Camera, using a Zeiss Colibri7 light source. Then, organoids were removed from the wells and transferred into tubes with OCT: 30% sucrose in PBS (2:1 v/v) O/N at 4 °C.

### Immunofluorescence

The tissues were sectioned with a cryostat (Leica CM3050S). Ten micrometers thick sections were placed on poly-lysine slides (Thermo Scientific, J2800AMNZ), kept at room temperature (RT) for 3 h with silica gel (Merck, 101969) to completely dry and then stored at −80 °C until use. All antibodies are described in Supplementary Data 11. All secondary antibodies were used at a dilution of 1:300–1:400.

For antigen retrieval (when necessary, see Supplementary Data 11), slides were placed in plastic jars with the appropriate solution and warmed at 80 °C for 30 min in a water bath. Then, the jars were placed in ice for 30 min to cool. Blocking was done with 5% donkey serum (Jackson Immuno-research, 017-000-121) for 1 h at RT and the primary antibodies were incubated at 4 °C O/N. After washes, the secondary antibodies were applied on the sections at RT for 1 h in the dark. The nuclei were counterstained with a DAPI solution 0.5 μg/μl (Biolegend, 422801) in PBS 1X Triton-X100 0.1%, and for mounting we used the ProLong Gold Antifade Reagent (Thermo-Fischer Scientific, P36934).

In the staining for cells that escaped inactivation of Fgfr2, we used extended antigen retrieval incubation (90 min) and employed a Biotin-Streptavidin staining strategy to improve the FGFR2 signal. In short, we used the Avidin/Biotin Blocking Kit (Vector Laboratories, SP-2001) after blocking and before primary antibody incubation, according to the manufacturer's suggestions (15 min Avidin, Rinse, 15 min Biotin, Rinse). After O/N incubation with primary antibodies and washes, we incubated the sections with a Biotin-SP-conjugated donkey anti-rabbit IgG (Secondary antibody) for 1 h at RT. After three washes, the sections were incubated with an Alexa Fluor® 647-conjugated Streptavidin for another 1 h at RT.

Images were acquired with Zeiss LSM780, LSM800 confocal microscopes (Carl Zeiss Microscopy GmbH, Jena, Germany) and a Zeiss Axio Observer Z.2 fluorescent microscope with Colibri2 or Colibri7, equipped with a Zeiss AxioCam 506 Mono digital camera and an automated stage. Image manipulations were with Fiji[113] and Zeiss Zen Blue 2.5.

## Quantification of S1 and S2 markers along the PD-axis

To quantify S1 or S2 marker co-expression, after immunofluorescence, we acquired five confocal microscopy images from P, I1-3 and D domains, from one P60 mouse lung section. Cell counting was performed using a custom pipeline at Cell Profiler 3.1. "*Global*" threshold strategy, "*otsu*" threshold method and three classes of thresholding were used. We manually curated the results for false positive and negative cells.

## Quantification of S1 and S2 markers in SPF and GF mice

For the quantification of immunofluorescence mean fluorescence intensity (MFI) of S1 and S2 markers in the left lung tissue sections of germ-free (GF) (P62) and specific pathogen-free (SPF) mice (P60), we manually defined Regions of Interest (ROIs) in proximal airways and TBs, using E-Cadherin and DAPI. Then, we measured their MFI with Zen Blue 2.5. The signal for the individual markers was normalized with the signal of E-Cadherin for each ROI, and the results from 3 animals per condition (SPF or GF) were combined. Statistical analysis for differences between proximal and distal airways was performed with a two-way unpaired T-test in GraphPad Prism.

## Scgb1a1 and Sftpb quantification in the adult Fgfr2-KO mice

We performed immunofluorescence for the indicated markers and E-Cadherin as described above. After image acquisition, we used aicspylibczi 3.2 to open and manipulate the *.czi files. We performed image registration with pystackreg 0.2.7[114]. We used E-Cadherin and DAPI for automatic cell segmentation with the graphical user interface of Cellpose 3.0.11[115]. Then, we measured the MFI within the segmented cells with geopandas 1.0.1[116]. We used Tissuumaps 3.2.1.11[117] to visually inspect and define the segmented cells in proximal and distal airway regions. For statistical analyses, we used the Microsoft Office Excel AVERAGE function to calculate arithmetic means for all proximal and all distal cells analyzed for each mouse. For visualization, values were plotted as bar plots with GraphPad Prism. Unpaired, two-tailed t-test was performed with GraphPad Prism.

## SCRINSHOT spatial analyses

For spatial analysis of the identified cell-types, we applied SCRINSHOT[64]. The padlock and detection probes are summarized in Supplementary Data 12. For image acquisition we used the same microscope setup as for immunofluorescence.

We used DAPI to align the images of the same areas between the hybridizations. We created multi-channel *.czi files with the signal of each detected gene as a unique channel and exported them as images (16-bit *.tiff format) using Zen Blue 2.5 (Carl Zeiss Microscopy, GmbH). Images were tiled with Matlab with Image Analysis toolbox (The MathWorks, Inc.). Manual nuclear segmentation was done with Fiji ROI Manager[113] and signal-dot counting was performed with Cell-Profiler 3.15[118]. Annotation of signal dots to the cells (2 μm expanded nuclei) was done with Fiji. For SCRINSHOT analyses of the feeder-free organoids and the adult Fgfr2-KO lungs, we performed image registration with pystackreg 0.2.7[114] and detected RCA products with BigFISH 0.6.2[119]. Cellpose 3.0.11[115] was used to automatically segment organoid cells (based on DAPI and RFP) and adult lung nuclei. We used geopandas 1.0.1 to register the detected mRNAs to the organoid cells and the expanded adult lung nuclei (for 2 μm). Further analyses were performed in R with custom scripts.

## S1 and S2 cell SCRINSHOT analysis

For the spatial analysis of S1- and S2-cells, we targeted the module-5 secreted proteins *Scgb3a1, Reg3g, Bpifb1, Tff2, Muc5b, Lgr6*, and *Pax9*, which are enriched in S1-cells. For the distal lung compartment (DP, S2, and alveolar cells), we used the module-2 markers *Hp* and *Scgb1a1* in addition to the module-3 surfactant proteins *Sftpc* and *Sftpb*, the enzyme *Atp8a1*, the IGF-signaling regulator *Igbp6* and the advanced glycosylation end products receptor *Ager*. We also targeted the ciliated-cell markers *Foxj1* and *Tuba1a* to recognize ciliated cells.

We analyzed P, I1, I2, I3, and D domains from three P60 mouse lungs. Nuclei were manually segmented in the acquired images based on DAPI and manually curated based on E-cadherin antibody staining, resulting to 6915 nuclei. Nuclear ROIs for each animal were expanded and filtered, keeping those with size between the mean cell-ROI size ± 2 standard deviations. Cell-ROIs with dots for only 1 analyzed marker were removed. We further filtered the cell-ROIs, keeping those with a total number of dots between the Mean number of dots ± 2 standard deviations. Cell-ROIs from all images were merged and log2-transformed [log2(dots + 1)]. After principal component analysis (PCA), the top up- or down-regulated genes of the first two principal components were used to cluster the Cell-ROIs with clusterboot, using the ward.D2 method. The heatmap of the analyzed cells was done with pheatmap-package in R. The balloon-plots of the expression levels (color intensity) and the percent of positive cells (size) were produced with ggpubr-package in R.

## DP-cell SCRINSHOT analysis

The gene-panel for the spatial analysis of DP-cells was based on differential expression analysis between S2-cells and AEC2a. The panel included *Scgb1a1* and *Sftpc* (the positivity of both defines the DP-cells/BASCs[26,28,29]) and the cytochrome genes *Cyp2f2* and *Cyp4b1* that showed high expression in S1- and S2-cells, moderate in DP and low in AEC2 cells. We additionally selected the secreted proteins *Lyz2* and *Lgi3*, the extracellular matrix proteins *Egfl6, Npnt* and *Col4a2*, the enzyme *Napsa*, the surface molecules *Cd74* and *Cldn18*, the transcription factors *Etv5* and *Rbpjl* and the negative regulator of Wnt-signaling *Nkd1*. We used *Ager* as a distal epithelial marker and the NE-cell markers, *Ascl1*, and *Calca* (Cgrp). In two independent experiments, we analyzed several lung areas from three adult (P60) mice and manually segmented 58072 nuclei. We kept cells with size between Mean cell-ROI size ± 2 standard deviations. We selected cell-ROIs with at least 40 dots of *Scgb1a1* plus *Sftpc* dots, in any combination and kept those with *Scgb1a1*-dots ≥ 5 or *Sftpc*-dots ≥ 5. The balloon-plots of the expression levels (color intensity) and the percent of positive Cell-ROIs (size) were produced with ggpubr-package[120] in R.

## SCRINSHOT analysis of the feeder-free organoids and adult Fgfr2-KO mice

We used Tissuumaps 3.2.1.11[117] to visually inspect and define the segmented cells within Cyp2f2high organoids in vitro and within proximal, intermediate and distal airway regions in vivo. SCRINSHOT count-matrices were further processed with Seurat. For the organoid experiments, we filtered out cells with fewer than 4 detected markers and 11 detected mRNAs. For the in vivo experiments, we removed mutant-lung cells with non-zero counts for the floxed Fgfr2 sequence and filtered out cells with less than 2 detected markers and 8 detected mRNAs. In both types of experiments, we randomly selected equal numbers of cells per condition. For clustering and UMAP-plot creation of the in vivo dataset, we applied SCTranform regularization and PCA analysis using the first eight PCs. We performed statistical analyses with MAST, using the log2-transformed (SCRINSHOT dots+1). We analyzed two independent feeder-free organoid experiments and three adult mice per condition in three independent experiments. For the SCRINSHOT experiments,

## Statistics and reproducibility

Statistical analyses of the results were done with a two-way multiple comparisons test in GraphPad Prism (GraphPad Software, Inc.) or by MAST in Seurat. In GraphPad Prism, for multiple comparisons, we used one-way and two-way ANOVA with Tukey's or Šídák's multiple comparisons tests, respectively. For pairwise comparisons, we used an unpaired, two-tailed t-test. Adjusted $p$-values in MAST were calculated with Bonferroni correction. No statistical method was used to predetermine sample size. No data were excluded from the analyses. The experiments were not randomized, and the Investigators were not blinded to allocation during experiments and outcome assessment.

## Reporting summary

Further information on research design is available in the Nature Portfolio Reporting Summary linked to this article.

## Data availability

Single-cell RNA-Seq data are available in GEO (lineage-tracing dataset of Scgb1a1-CreER$^{T2\ pos/neg}$; Rosa26-fGFP$^{pos/neg}$ cells: GSE215957, adult dataset of Scgb1a1-CreER$^{T2\ pos/neg}$; Rosa26-Ai14$^{pos/neg}$ cells: GSE216210 and Fgfr2-inactivation dataset of Scgb1a1-CreER$^{T2pos/neg}$; Rosa26-Ai14$^{pos/pos}$; Fgfr2$^{fl/fl}$ cells: GSE216451). Scripts and RAW-image data and the analysis results can be found in Zenodo (https://doi.org/10.5281/zenodo.10418253 [https://zenodo.org/records/10418253] and https://doi.org/10.5281/zenodo.17789722 [https://zenodo.org/records/17789722]) and/or within the Source data files. Source data are provided with this paper.

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

## Acknowledgements

We thank the SciLifeLab NGI and WABI for long-term bioinformatics support. We thank the Karolinska Institute Biomedicum Flow Cytometry Core Facility (BFC) for the FACS-sorting services. We acknowledge resources provided by the Swedish National Infrastructure for Computing (SNIC) at UPPMAX, partially funded by the Swedish Research Council through grant agreement no. 2018-05973 (projects b2015134 and SNIC 2021/22-431). We thank the Imaging Facility, Stockholm

University (IFSU), and the Stockholm University Experimental Core Facility (ECF) for the excellent technical support. The work was supported by grants from VR 2023-03087 and 2019-04893, CF 243852 Pj 01H and the Erling Persson Foundation 2023-0035 to C.S., Knut and Alice Wallenberg Foundation (2020.0057) and the VR 2021- 04896 to K.G. We acknowledge the European Respiratory Society-EMBO for the European Respiratory Society Long-Term Research Fellowship (Reference Number: LTRF 2014 – 3565) and the Royal Swedish Academy of Sciences for the research grant (BS2025-0017) to A.S.

## Author contributions

A.S. performed and analyzed experiments with the adult lungs, lineage-tracing, and feeder-free organoid experiments. A.L. performed the spatial analyses of the developing and adult lungs. J.T. performed the spatial analyses of the germ-free mice and contributed to the analyses of the feeder-free organoids and the Fgfr2-KO mice together with A.K. A.B.F. performed and analyzed the organoid co-culture experiments. Å.B. contributed to the scRNA-Seq analysis of the lineage-tracing experiments. O.E. contributed to the analysis of the feeder-free organoids. A.F., P.M.B., and E.B. performed crosses for the in vivo Fgfr2-inactivation experiments. J.K. performed the FACS-sorting and scRNA-Seq library preparations of the Fgfr2-KO mouse experiments. L.M.H. and F.B. contributed germ-free mice. K.G. and C.B. contributed to the lineage-tracing experiments. W.S. and S.B. contributed mice and resources. C.S. designed and supervised the project together with A.S. All authors contributed to the manuscript preparation and writing.

## Funding

## Competing interests

The authors declare no competing interests.
