## [Transparent Peer Review File · Nature Communications]

FGFR-signaling establishes spatial gradients of secretory cell identities along the airway proximal-distal axis

Corresponding Author: Dr Alexandros Sountoulidis

Version 0:

Reviewer comments:

Reviewer #1

(Remarks to the Author)

The primary new data in the revision is from the adult *Fgfr2* deletion which provides new information, although similar to what has been observed in AT2 cells. Also the enrichment of ribosomal and mitochondrial gene signatures in the *Fgfr2* KO suggest cell stress-have the authors looked at whether this is the main phenotype in this KO model? The refusal to drop the "secretory" nomenclature for AT2 cells is simply wrong-there is no need to relabel these well established cells based on little to no new information. While the new single cell data is interesting, it really does not extend much beyond what has already been reported by this group and others.

Reviewer #2

(Remarks to the Author)

This is an excellent manuscript, further improved by the new experiments and the rewriting has made it much easier to read.

Reviewer #3

(Remarks to the Author)

All my concerns were addressed either with addition of new data or provided explanation.

Response to Referees Letter

Reviewers' comments:

Reviewer #1 (Remarks to the Author):

The primary new data in the revision is from the adult *Fgfr2* deletion which provides new information, although similar to what has been observed in AT2 cells.

Fgfr2 signaling prevents AT2 apoptosis in adults. Shortly after *Fgfr2*-inactivation, AT2 cells undergo apoptosis and do not show signs of converting to AT1 cells (PMID: 36414616). However, after longer periods upon the inactivation *Fgfr2* becomes dispensable for adult AT2 homeostasis, presumably because of compensatory regeneration (analysis 1 month, 6 months, and 1 year post *Fgfr2*-inactivation) (PMID: 33979629). Our results show that in the airway epithelium *Fgfr2* is required for the expression of *Scgb1a1* and *Sftpb* and for the maintenance of DP-cells (BASCs). There is also a minor induction of the S1 markers but AT1 marker expression remains unaffected. This indicates that there is no similarity between the defects caused by *Fgfr2*-inactivation in AT2 and in airway secretory cells.

Also the enrichment of ribosomal and mitochondrial gene signatures in the *Fgfr2* KO suggest cell stress—have the authors looked at whether this is the main phenotype in this KO model?

We have shown that the perinatal *Fgfr2*-inactivation in the airways upregulates nuclear genes encoding mitochondrial components, sodium channel genes (*Commd3*, *Scnn1a*, *Scnn1b*) and genes related to autophagy and vesicle trafficking (*Creg1*, *Vamp8*, *Vamp5*, *Rab25* and *Rabac*), which are normally expressed by AT1 cells (Fig.5C, Suppl. Fig. 5D, Suppl. Fig. 6A-C). In addition, perinatal *Fgfr2* mutant cells fail to suppress the expression of genes encoding ribosomal proteins, which are also normally downregulated during airway maturation. In addition, and only in the intermediate airway regions, *Fgfr2* mutant cells failed to suppress various perinatally upregulated genes encoding ECM proteins and innate immunity genes. The extensive comparison of up- and down-regulated gene expression programs during normal airway maturation to the ones affected by *Fgfr2*-inactivation indicates that FGFR2 controls airway epithelial maturation and differentiation without interfering with “stress-related” genes.